# The molecular basis for ANE syndrome revealed by the large ribosomal subunit processome interactome

Kathleen L McCann[1], Takamasa Teramoto[2], Jun Zhang[2], Traci M Tanaka Hall[2]*, Susan J Baserga[1,3,4]*

[1]Department of Genetics, Yale University School of Medicine, New Haven, United States; [2]Epigenetics and Stem Cell Biology Laboratory, National Institute of Environmental Health Sciences, National Institutes of Health, Research Triangle Park, United States; [3]Department of Therapeutic Radiology, Yale University School of Medicine, New Haven, United States; [4]Department of Molecular Biophysics and Biochemistry, Yale University School of Medicine, New Haven, United States

**Abstract** ANE syndrome is a ribosomopathy caused by a mutation in an RNA recognition motif of RBM28, a nucleolar protein conserved to yeast (Nop4). While patients with ANE syndrome have fewer mature ribosomes, it is unclear how this mutation disrupts ribosome assembly. Here we use yeast as a model system and show that the mutation confers growth and pre-rRNA processing defects. Recently, we found that Nop4 is a hub protein in the nucleolar large subunit (LSU) processome interactome. Here we demonstrate that the ANE syndrome mutation disrupts Nop4's hub function by abrogating several of Nop4's protein-protein interactions. Circular dichroism and NMR demonstrate that the ANE syndrome mutation in RRM3 of human RBM28 disrupts domain folding. We conclude that the ANE syndrome mutation generates defective protein folding which abrogates protein-protein interactions and causes faulty pre-LSU rRNA processing, thus revealing one aspect of the molecular basis of this human disease.

*For correspondence: hall4@ niehs.nih.gov (TMTH); susan. baserga@yale.edu (SJB)

**Competing interests:** The authors declare that no competing interests exist.

## Introduction

Ribosomes are essential for life. The fundamental cellular process of ribosome assembly requires the coordinated action of all three RNA polymerases, over 200 biogenesis factors, and a number of small nucleolar RNAs (*Thomson et al., 2013*; *Woolford and Baserga, 2013*; *Fernández-Pevida et al., 2015*). In yeast, ribosome biogenesis initiates in the nucleolus with the transcription of the 35S poly-cistronic pre-ribosomal RNA (rRNA) precursor by RNA polymerase I. The 35S pre-rRNA undergoes a number of cleavage and modification events to give rise to the mature 18S, 5.8S and 25S rRNAs. Mutations that partially disrupt ribosome assembly or function are often deleterious and can lead to disease in humans. Collectively, these diseases of ribosome biogenesis are called ribosomopathies. Ribosomopathies are caused by mutations in proteins that function in all stages of ribosome assembly (*McCann and Baserga, 2013*; *Armistead and Triggs-Raine, 2014*).

A homozygous missense mutation in the nucleolar protein RBM28 causes the ribosomopathy, alopecia, neurological defects and endocrinopathy (ANE) syndrome (*Nousbeck et al., 2008*). Five affected children of a consanguineous kindred displayed degrees of baldness, mental retardation, motor deterioration, and reduced pituitary gland function in their second decade of life (*Nousbeck et al., 2008*; *Warshauer et al., 2015*). The mutation segregated in the family in an autosomal recessive manner and was mapped to a single leucine to proline amino acid substitution at position 351 (L351P) of RBM28. This residue resides in the first α-helix of its third RNA recognition

**eLife digest** ANE syndrome is a rare genetic disease that causes many problems including hair loss, mental retardation and a failure to develop normally during puberty. A study of 5 boys in the same family that were all born with the condition revealed that the disease is caused by a small change (or mutation) in a protein called RBM28. While little is known about the role of human RBM28, it is known that the equivalent protein in yeast – known as Nop4 – plays a critical role in forming a network of proteins needed to assemble ribosomes, the machines that make proteins.

McCann et al. investigated how such a small mutation in human RBM28 could cause disease and whether this involves interrupting the assembly of ribosomes. The experiments show that introducing the same mutation into yeast Nop4 impaired the ability of Nop4 to form the network of proteins needed for ribosomes to assemble. This ultimately restricted the growth of the yeast.

Further experiments revealed that the mutation also alters the shape of the human RBM28 protein. The main challenges for the future are to find out whether human RBM28 plays a similar role in ribosome assembly as the yeast protein, and to work out how disrupting ribosome assembly could lead to the symptoms of ANE syndrome.

motif (RRM3; *Figure 1A*, *Figure 1—figure supplement 1*). ANE syndrome was classified as a ribosomopathy because RBM28 is localized to the nucleolus and because patient fibroblasts showed reduced numbers of ribosomes (*Damianov et al., 2006*; *Nousbeck et al., 2008*). While the L351P mutation is predicted to disrupt the first α-helix of RRM3 and thereby impair RBM28 function, it is not known how this single amino acid substitution disrupts the normal function of RBM28 in the nucleolus. Specifically, what is the molecular basis of ANE syndrome pathogenesis?

Nop4, the yeast ortholog of RBM28, is required for assembly of the large ribosomal subunit (LSU; *Bergès et al., 1994*; *Sun and Woolford, 1994*; *Nousbeck et al., 2008*). Recently, the LSU processome interactome revealed that Nop4 functions as a hub protein, interacting with many more proteins than average within the LSU processome (*McCann et al., 2015*). We hypothesized that introduction of the orthologous ANE syndrome mutation into Nop4 (L306P) would disrupt Nop4's function as a hub protein and therefore disrupt LSU assembly in the nucleolus.

We demonstrate that introduction of the ANE syndrome mutation into Nop4 disrupts growth and pre-rRNA processing in yeast and abrogates several, but not all, of its protein-protein interactions. Surprisingly, the C-terminal half of Nop4, where the ANE syndrome mutation occurs, is necessary and sufficient for hub protein function, cell growth and pre-rRNA processing. Consistent with these findings, circular dichroism and NMR reveal that the ANE mutation in RRM3 of the Nop4 human ortholog, RBM28, disrupts folding of the entire domain, not just the first α-helix. Together, these results suggest that the molecular basis of ANE syndrome lies in defective protein folding that reduces protein interactions and the function of RBM28 as a hub protein, resulting in pre-rRNA processing defects in the nucleolus.

## Results

### The ANE syndrome mutation in Nop4 causes growth defects in yeast

Our goal was to elucidate the molecular basis of the ribosomopathy ANE syndrome, which is attributed to a single amino acid substitution, L351P, in RBM28 (*Nousbeck et al., 2008*). Human RBM28 can complement the growth defect in the yeast, *Saccharomyces cerevisiae*, when its ortholog, the essential Nop4 protein, is depleted (*Figure 1—figure supplement 2*; *Kachroo et al., 2015*). Therefore, we used yeast genetics to pinpoint the molecular basis of ANE syndrome. A ClustalX alignment of the yeast Nop4 and human RBM28 amino acid sequences permitted identification of the orthologous ANE syndrome mutation in Nop4 (*Figure 1—figure supplement 3*). Over their entire length, the amino acid sequences of Nop4 and RBM28 are ~26% identical and 34% similar, and both contain four RRMs (*Figure 1A*). The ANE syndrome mutation in human RBM28, L351P, is within the third RRM. Inspection of the alignment revealed an orthologous leucine in the third RRM of yeast Nop4,

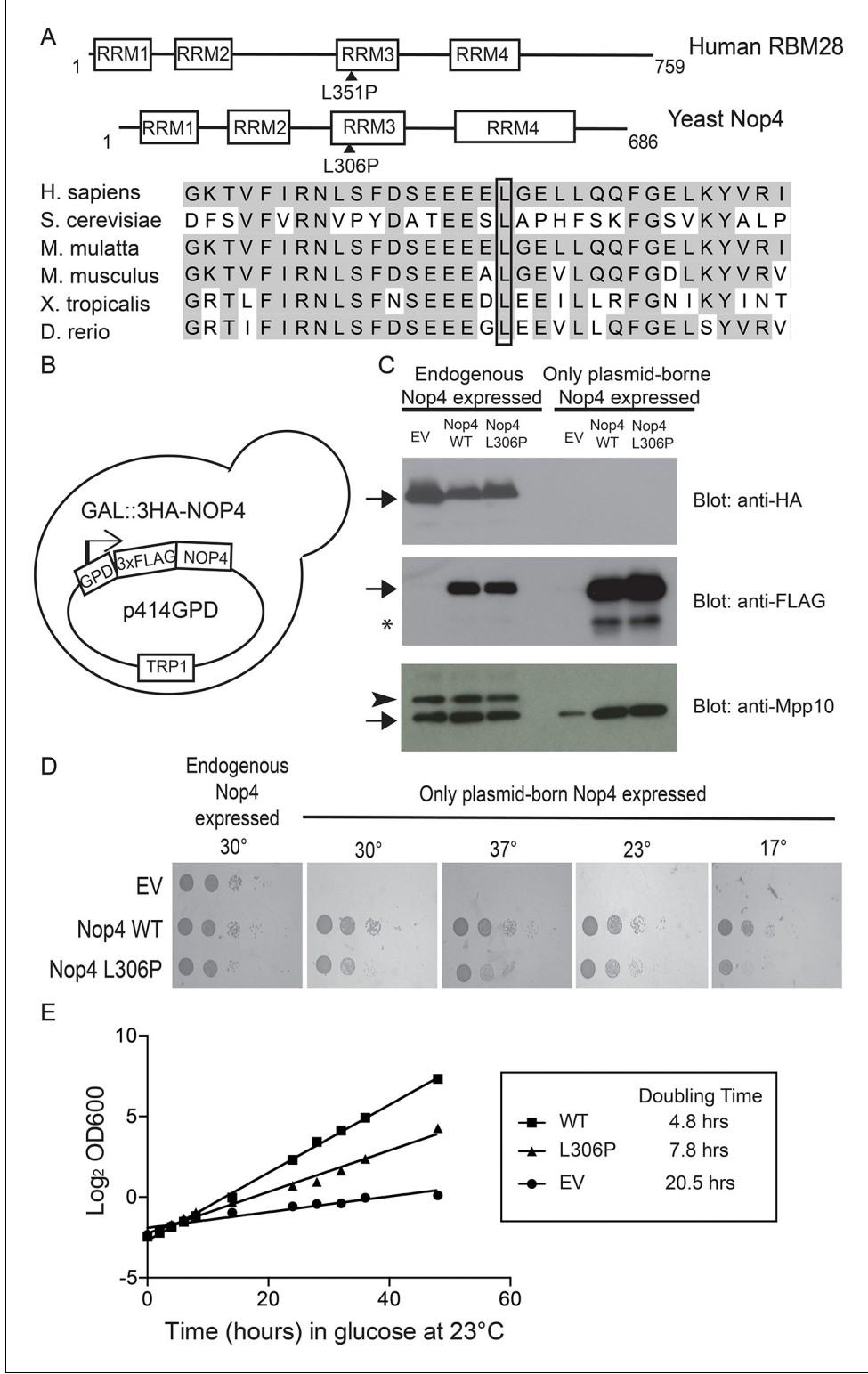

**Figure 1.** The ANE syndrome mutation confers a growth defect in yeast. (**A**) The leucine that is mutated in ANE syndrome is highly conserved. Top: Diagram of the domain structure for human RBM28 and its yeast ortholog, Nop4. The boxes represent RNA Recognition Motifs (RRMs). Arrowheads indicate the approximate location of the mutated amino acid, L351P in humans and L306P in yeast. Bottom: Multiple sequence alignment of the portion of RRM3 containing the mutated leucine. Shaded amino acids in are conserved. A box outlines the conserved leucine that is mutated in ANE syndrome. (**B**) Schematic of the yeast strain used for testing the ANE syndrome mutation in

*Figure 1 continued on next page*

*Figure 1 continued*

Nop4. Endogenous Nop4 was placed under the control of the inducible *GAL4* promoter in haploid yeast. FLAG-tagged unmutated Nop4 WT or Nop4 L306P was expressed constitutively from the p414GPD plasmid. (**C**) Nop4 WT and Nop4 L306P are expressed at equivalent levels from the yeast expression vector p414GPD-3xFLAG-GW. The depletion of endogenous Nop4 protein was confirmed by western blot using an HRP-conjugated monoclonal antibody against the 3xHA tag. Expression of Nop4 WT or Nop4 L306P from p414GPD-3xFLAG-GW was analyzed by western blot using a monoclonal antibody against the 3xFLAG tag. As a loading control, a western blot using α-Mpp10 was performed. The expression levels of Nop4 WT and Nop4 L306P relative to Mpp10 were quantitated and normalized to Nop4 WT: Nop4 WT = 1, Nop4 L306P = 0.96. EV = empty vector. The arrows indicate the expected bands. The arrowhead indicates an Mpp10 species only observed when yeast are grown in galactose and raffinose. The asterisk denotes degradation. (**D**) The ANE syndrome mutation in Nop4 impairs growth on solid medium. Serial dilutions of yeast expressing the indicated Nop4 constructs were grown on solid medium for 3 days at 30°C and 37°C or for 5 days at 23°C and 17°C. Three biological replicates were performed starting with transformation of the plasmids into the yeast strain. (**E**) The Nop4 ANE syndrome mutation impairs growth in liquid medium. Yeast expressing the indicated Nop4 constructs were transferred from SG/R-Trp to SD-Trp and 23°C to deplete the endogenous Nop4. Growth was monitored for 48 hr by measuring the absorbance at $OD_{600}$. The $\log_2$ of the $OD_{600}$ was plotted over time and the slope was used to estimate the doubling time. Four biological replicates were performed starting with transformation of the plasmids into the yeast strain.

The following figure supplements are available for figure 1:

**Figure supplement 1.** Multiple sequence alignment of RRM3 from RBM28.

**Figure supplement 2.** RBM28 complements the growth defect in yeast observed upon depletion of its essential ortholog, Nop4.

**Figure supplement 3.** Amino acid sequence alignment of human RBM28 and its yeast ortholog Nop4.

which we mutated to proline to introduce the ANE syndrome mutation into Nop4 (L306P; *Figure 1A*; *Figure 1—figure supplement 1*).

We determined that the ANE syndrome mutation in Nop4 (L306P) impaired yeast growth. We generated a strain where endogenous *NOP4* is under the control of a galactose-inducible, glucose-repressible promoter and tagged with a triple-HA epitope (*Figure 1B*). Unmutated (wild type; WT) Nop4 or Nop4 L306P protein is tagged with a triple-FLAG epitope and constitutively expressed from a plasmid (p414GPD). Western blotting of total protein demonstrated that after growth of this strain in glucose for 48 hr at 23°C, the endogenous Nop4 was reduced to undetectable levels and plasmid-borne Nop4 WT and Nop4 L306P were expressed at comparable levels (*Figure 1C*). Serial dilutions of strains bearing the plasmids: empty vector (EV), Nop4 WT and Nop4 L306P were spotted onto plates containing glucose and incubated at 30°C, 37°C, 23°C and 17°C. At all tested temperatures, depletion of Nop4 (EV) conferred a severe growth defect relative to growth of Nop4 WT (*Figure 1D*). The L306P mutation impaired growth at all temperatures tested compared to WT, although the defect was not as severe as that observed with the EV control (*Figure 1D*).

To confirm our findings, we analyzed growth in liquid medium at 23°C and estimated the doubling time for each strain. Endogenous Nop4 was depleted and the growth of strains bearing the plasmids: EV, Nop4 WT or Nop4 L306P was monitored for 48 hr. Similar to growth on solid medium, Nop4 L306P exhibited a moderate growth defect in liquid culture, doubling every 7.8 hr, compared to WT, which doubled every 4.8 hr; however, the defect was not as severe as that observed with the EV control, which doubled every 20.5 hr (*Figure 1E*).

## The ANE syndrome mutation causes pre-rRNA processing defects in yeast

The ANE syndrome mutation, L306P in yeast Nop4, also disrupts pre-rRNA processing. As growth defects caused by mutation of a nucleolar protein are often indicative of ribosome biogenesis defects, we tested whether the growth defects conferred by Nop4 L306P were due to disruption of ribosome biogenesis. Previously, it has been shown that the mature 25S rRNA and the 27S and 7S pre-rRNA precursors are severely reduced in yeast depleted of Nop4 (*Figure 2A*; *Bergès et al.,*

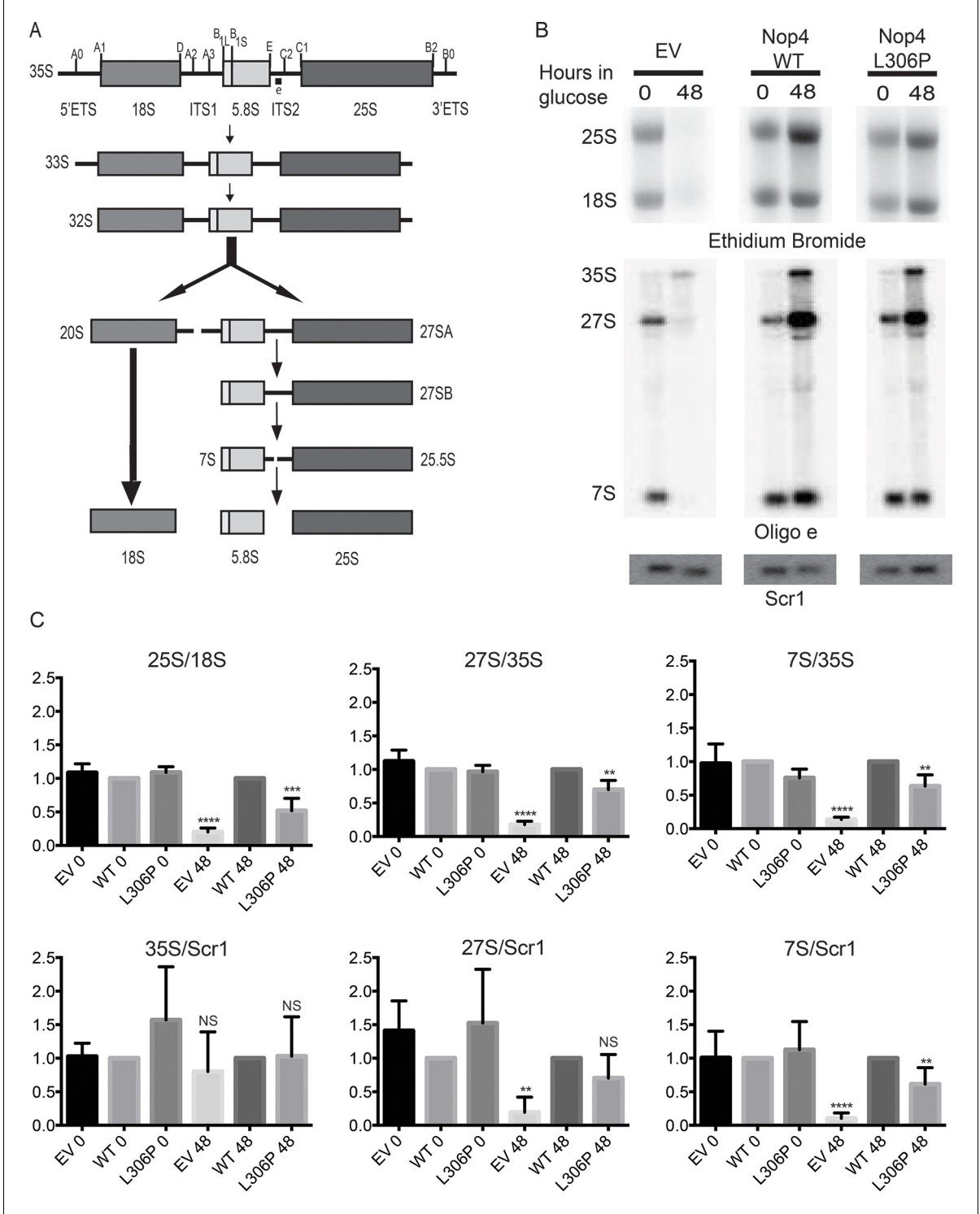

**Figure 2.** The ANE syndrome mutation disrupts pre-rRNA processing in yeast. (**A**) Simplified diagram depicting the pre-rRNA processing steps in yeast. The pre-rRNA is transcribed as a 35S polycistronic precursor. The external transcribed spacers (5′ and 3′ ETS) and the internal transcribed spacers (ITS1 and 2) are removed through a number of cleavage steps to produce the mature 18S, 5.8S and 25S rRNAs. Oligonucleotide probe e, which is complementary to ITS2 and detects all 27S and 7S pre-rRNAs (indicated on top line), was used for northern blotting. (**B**) The ANE syndrome mutation in Nop4 impairs pre-rRNA processing in yeast. Top panel: Ethidium bromide staining of total RNA extracted from yeast expressing no Nop4 (empty vector; EV), Nop4 WT or Nop4 L306P after depletion of endogenous Nop4 for the indicated time. Bottom panel: Northern blots of total RNA using radio-labeled oligonucleotide probe e to detect 35S, 27S, and 7S pre-rRNAs and an oligonucleotide probe complementary to Scr1 as a loading control. (**C**) The ratios of the mature rRNAs (25S/18S), the ratios of the precursors (27S/35S and 7S/35S) and the ratios of the precursors to the loading control

*Figure 2 continued on next page*

*Figure 2 continued*

Scr1 (35S/Scr1, 27S/Scr1 and 7S/Scr1) were calculated from four replicate experiments and were plotted with error bars representing the standard deviation. The significance of the ratios of Nop4 depleted yeast (empty vector; EV) or Nop4 L306P compared to WT was evaluated using one-way ANOVA. ****indicates a p value < 0.0001. ***indicates a p value < 0.001. **indicates a p value <0.01. NS = not significant. Four biological replicates were performed.

The following source data is available for figure 2:

**Source data 1.** Quantitation and statistical analyses for *Figure 2C*.

*1994*; *Sun and Woolford, 1994*). To determine whether Nop4 L306P similarly disrupts production of the 25S rRNA, total RNA was harvested from strains bearing plasmids expressing no Nop4 (empty vector; EV), Nop4 WT or Nop4 L306P and depleted of endogenous Nop4 for 0 and 48 hr. The 25S and 18S rRNAs were visualized by ethidium bromide staining, quantified and the ratio of 25S/18S, a measure of the relative levels of the mature rRNAs, was calculated and normalized to Nop4 WT for each time point (*Figure 2B* top panels). The observed decrease in the 25S/18S ratios correlated with the trend of the observed growth defects. The EV control, which had the most severe growth defect, also had the most severe reduction in 25S/18S ratio levels in comparison to Nop4 WT. Nop4 L306P conferred a moderate growth defect and a moderate, but statistically significant, reduction in the 25S/18S rRNA ratio (*Figure 2C*), consistent with reduced 25S levels.

Since the L306P mutation resulted in a reduction of the 25S/18S ratio, we determined whether the L306P mutation had an effect on pre-rRNA processing. Northern blot analysis of total RNA harvested from strains expressing no Nop4 (EV), Nop4 WT or Nop4 L306P after depletion of endogenous Nop4 for 0 and 48 hr was performed using an oligonucleotide probe in ITS2 and an oligonucleotide probe against the loading control Scr1 (*Figure 2A*). The ratios of 27S/35S and 7S/35S pre-rRNAs as well as the ratios of the precursors to the loading control, Scr1, were quantified and normalized to Nop4 WT. Similar to the 25S/18S ratios, the pre-rRNA processing defects mirror the growth defects. Depletion of Nop4 (EV) resulted in a severe reduction of 27S and 7S levels, with a concomitant decrease in the 27S/35S, 7S/35S, 27S/Scr1 and 7S/Scr1 ratios, indicative of an ITS1 processing defect, as has been previously observed (*Figure 2B,C*; *Bergès et al., 1994*; *Sun and Woolford, 1994*). The Nop4 L306P mutant showed an intermediate growth defect and also displayed an intermediate, but statistically significant, ITS1 processing defect as indicated by reduced 27S/35S, 7S/35S and 7S/Scr1 ratios (*Figure 2B,C*).

## The ANE syndrome mutation disrupts Nop4 protein-protein interactions

As the LSU processome interactome revealed that Nop4 functions as a hub protein (*McCann et al., 2015*), we tested whether the ANE mutation in Nop4 abrogates protein-protein interactions using a directed yeast two-hybrid (Y2H) assay (*Figure 3A*). Nop4 WT and Nop4 L306P were expressed at comparable levels as prey fusion proteins from the Y2H vector, pACT2, in PJ69-4α (*Figure 3B*). Yeast expressing either of these prey proteins or no Nop4 (empty vector; EV) were co-transformed with the Y2H bait vector, pAS2-1, encoding 5 Nop4-interacting proteins (*Table 1*; *McCann et al., 2015*), including Nop4 itself, and tested for interaction by serial dilution on the indicated selective medium (*Figure 3C*).

The presence of the ANE syndrome mutation (L306P) disrupted the interaction between Nop4 and 4 of the 5 interacting partners we tested (*Figure 3C*), including Nop4 itself. While all 5 bait proteins interacted with Nop4 WT, as indicated by growth on SD-Leu-Trp-His + 6 mM 3-AT, Mak5, Nop4, and Nsa2 did not interact with Nop4 L306P, as no growth was observed (*Figure 3C*). Noc2 did interact with Nop4 L306P, but growth was reduced compared to WT. In contrast, Dbp10 interaction with Nop4 was unaffected by the L306P mutation (*Figure 3C*). Thus, the presence of the ANE syndrome mutation (L306P) disrupts Nop4 interaction with a subset of Nop4's interacting proteins (Mak5, Nop4, Nsa2 and Noc2) by Y2H, suggesting that the ANE syndrome mutation disrupts Nop4 function as a hub protein in the LSU processome.

To confirm our findings, we utilized a co-immunoprecipitation method developed to validate Y2H datasets (*McCann et al., 2015*) to assay for changes in protein-protein interactions in the presence

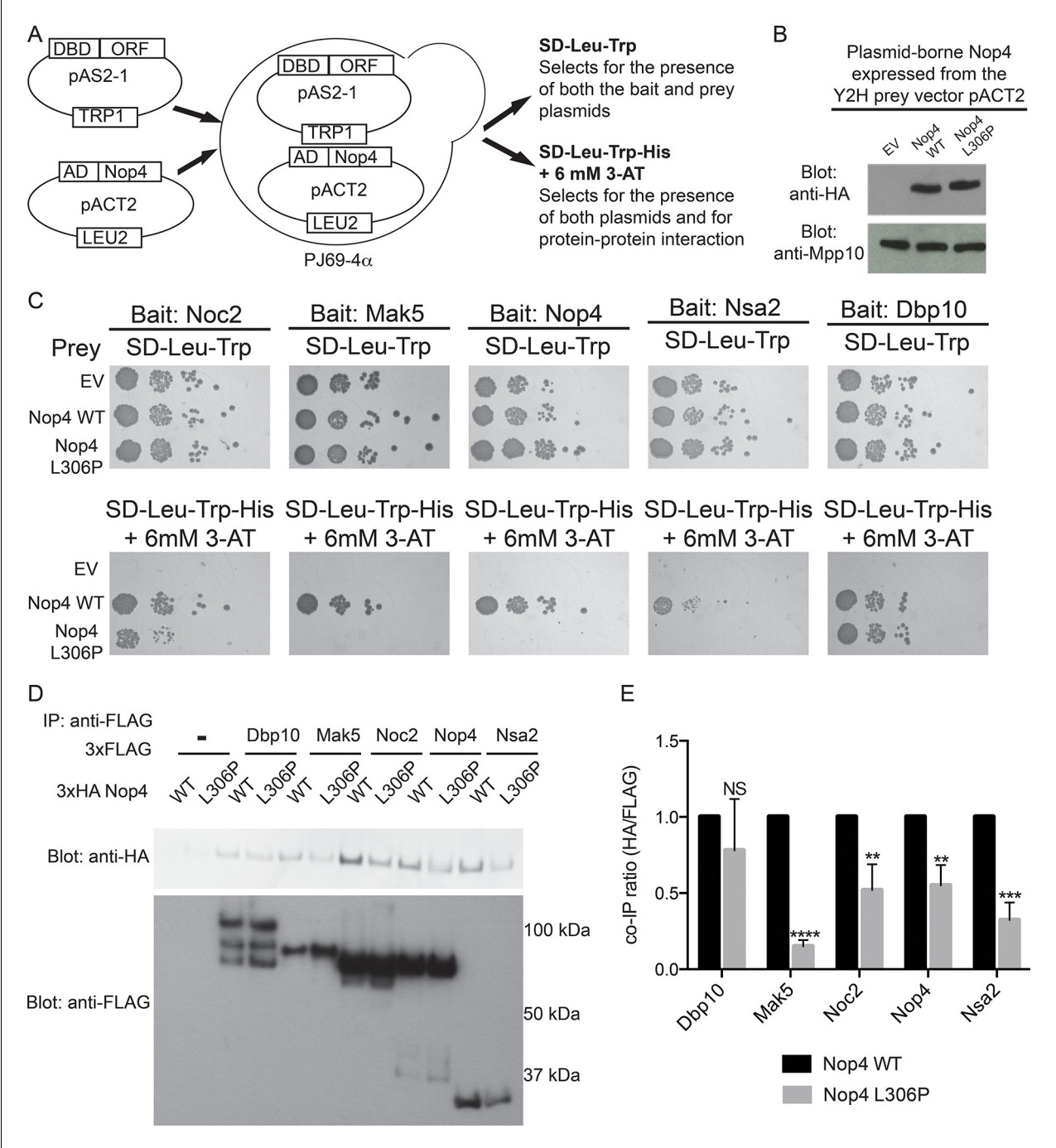

**Figure 3.** The ANE syndrome mutation in Nop4 disrupts protein-protein interactions. (**A**) Schematic of Y2H analysis. Nop4 WT or Nop4 L306P were cloned into the prey vector (pACT2) while five Nop4 interacting proteins (Noc2, Mak5, Nop4, Nsa2 and Dbp10) were cloned into the bait vector (pAS2-1). Each bait was individually co-transformed into the yeast strain PJ69-α with empty vector (EV), Nop4 WT or Nop4 L306P prey and spotted onto medium to confirm the presence of both Y2H plasmids (SD-Leu-Trp) and onto medium to test for protein-protein interactions (SD-Leu-Trp-His + 6 mM 3-AT). (**B**) Nop4 WT and Nop4 L306P are expressed at equivalent levels from the Y2H vector pACT2. Total protein was extracted from PJ69-4α yeast transformed with EV or expressing Nop4 WT or Nop4 L306P from the Y2H prey vector, pACT2. Nop4 WT and Nop4 L306P are expressed as fusions with the GAL4 activation domain and a 3xHA tag. Protein extracts were separated by SDS-PAGE and analyzed by α-HA western blot. As a loading control, a western blot using α-Mpp10 was performed. The expression levels of Nop4 WT and Nop4 L306P relative to Mpp10 were quantitated and normalized to Nop4 WT: Nop4 WT = 1, Nop4 L306P = 1.1 (**C**) Y2H analysis by serial dilution reveals that the ANE syndrome (L306P) mutation disrupts some Nop4 protein-protein interactions. Two biological replicates of a subset of interacting proteins were performed starting with co-transformation of the bait and prey plasmids into the Y2H strain. (**D**) The ANE syndrome (L306P) mutation reduces protein-protein interactions as determined by co-

*Figure 3 continued on next page*

*Figure 3 continued*

immunoprecipitation. Yeast extract was generated from yeast expressing either Nop4 WT or Nop4 L306P and one of its interacting partners and incubated with α-FLAG resin. Co-immunoprecipitations were assessed by α-HA western blot. The expected molecular weights of the Nop4 interacting proteins are: Dbp10 = 113 kDa, Mak5 = 87 kDa, Noc2 = 82 kDa, Nop4 = 78 kDa and Nsa2 = 30 kDa. (E) The ratio of co-purified 3xFLAG tagged Nop4 WT or Nop4 L306P to co-immunoprecipitated 3xHA tagged interacting partner was calculated from three replicate experiments and plotted with error bars representing the standard deviation. The significance of the co-immunoprecipitation ratio of Nop4 L306P compared to WT for each interacting partner was evaluated using a t-test. ****indicates a p value < 0.0001. ***indicates a p value < 0.001. **indicates a p value <0.01. NS = not significant. Three biological replicates were performed.

The following source data is available for figure 3:

**Source data 1.** Quantitation and statistical analyses for *Figure 3E*.

of the ANE syndrome mutation (L306P). Nop4 WT and Nop4 L306P were expressed as 3xHA fusion proteins from the modified yeast expression vector p414GPD-3xFLAG and the 5 Nop4-interacting proteins were expressed as 3xFLAG fusion proteins from the modified yeast expression vector p415GPD-3xHA (*Mumberg et al., 1995*; *McCann et al., 2015*). The plasmids were co-transformed into yeast, immunoprecipitations were performed with anti-FLAG resin and the co-purifying Nop4 proteins were visualized by Western blotting with an antibody to the 3xHA tag (*Figure 3D*). The ratio of co-purifying 3xHA-Nop4 or 3xHA-Nop4 L306P to co-immunoprecipitated 3xFLAG-

**Table 1.** Nop4 interacts with 23 large subunit assembly factors with high confidence. The Nop4 interacting proteins were identified by yeast two-hybrid and were assigned a confidence score in (*McCann et al., 2015*).

| Nop4 Interacting Partner | Confidence Score (from *McCann et al., 2015*) |
|---|---|
| Nop4 | 92% |
| Loc1 | 92% |
| Ebp2 | 85% |
| Nop12 | 85% |
| Nsa2 | 85% |
| Mak5 | 84% |
| Cgr1 | 70% |
| Cic1 | 70% |
| Has1 | 70% |
| Noc2 | 70% |
| Nop13 | 70% |
| Nsr1 | 70% |
| Rrp12 | 70% |
| Rrp14 | 70% |
| Mak21 | 68% |
| Dbp10 | 63% |
| Drs1 | 63% |
| Nop16 | 63% |
| Nug1 | 63% |
| Prp43 | 63% |
| Spb4 | 63% |
| Tma16 | 63% |
| Nog1 | 53% |

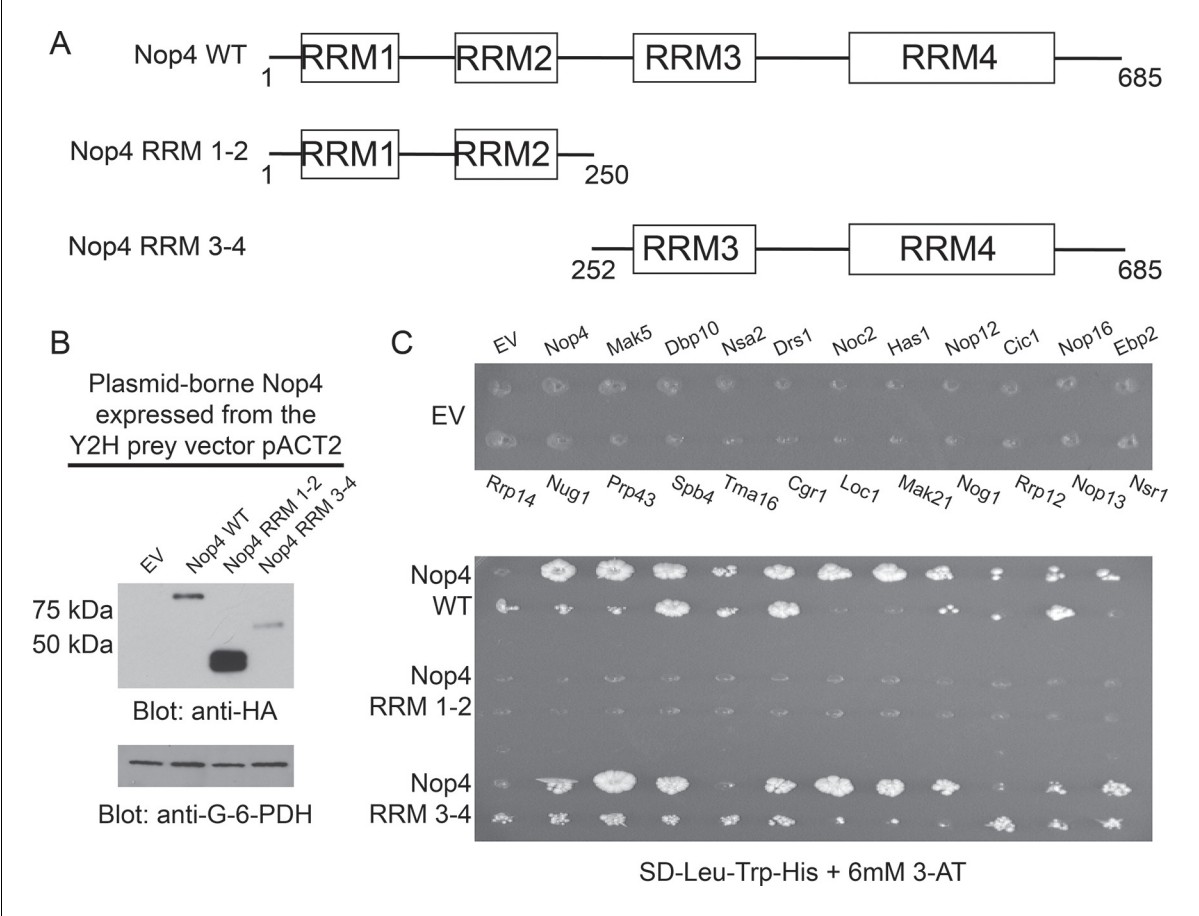

**Figure 4.** RRM3 and RRM4 of Nop4 mediate protein-protein interactions. (**A**) Schematic representation of Nop4 RRM domains and the N- and C-terminal fragments containing RRMs 1 and 2 (RRM 1–2; residues 1–250) or 3 and 4 (RRM 3–4; residues 252–685), respectively. (**B**) Nop4 WT and the Nop4 fragments are differentially expressed from the Y2H vector pACT2. Total protein was extracted from PJ69-4α yeast transformed with empty vector (EV) or expressing Nop4 WT (78 kDa), Nop4 RRM 1–2 (28.3 kDa) or Nop4 RRM 3–4 (49.4 kDa) from the Y2H prey vector, pACT2. Nop4 WT, Nop4 RRM 1–2 and Nop4 RRM 3–4 are expressed as fusions with the GAL4 activation domain and a 3xHA tag. Protein extracts were separated by SDS-PAGE and analyzed by α-HA western blot. As a loading control, a western blot using α-Glucose-6-Phosphate Dehydrogenase (G-6-PDH) was performed. The expression levels of Nop4 WT, Nop4 RRM 1–2 and Nop4 RRM 3–4 relative to G-6-PDH were quantitated and normalized to Nop4 WT: Nop4 WT = 1, Nop4 RRM 1–2 = 7.4, Nop4 RRM 3–4 = 0.29. (**C**) Y2H analysis demonstrates that Nop4 RRM 3–4 mediates protein-protein interactions. Nop4 WT and the Nop4 fragments described in (a) were tested as preys for interaction with 23 Nop4 interacting proteins as baits. The baits are labeled for the empty vector (EV) control plate. Growth on selective medium (SD-Leu-Trp-His + 6 mM 3-AT) indicates an interacting bait-prey pair. Two biological replicates were performed starting with the transformation of the bait and prey plasmids into the Y2H strains.

interacting protein was calculated and normalized to WT for each interacting partner (*Figure 3E*). Interestingly, 3xHA-Nop4 L306P co-purified significantly less efficiently than 3xHA-Nop4 WT with all interacting partners assayed, except Dbp10, as was observed by Y2H. Thus, the ANE syndrome mutation disrupts or reduces a subset of Nop4 protein-protein interactions.

## RRM3 and RRM4 of Nop4 mediate protein binding

We hypothesized that Nop4 RRM3 may be important for protein binding since it contains the ANE syndrome mutation (L306P) that, when present, abrogates interaction with a subset of Nop4 interacting proteins (*Figure 1A*; *Figure 3C–E*). Although RRM domains typically bind RNA, there are several published examples of RRMs that bind proteins rather than RNA (*Fribourg et al., 2003*; *Lau et al., 2003*; *Selenko et al., 2003*; *Bono et al., 2004*; *Kadlec et al., 2004*). To determine the contribution of the 4 RRMs to Nop4's function as a protein-binding hub, we divided Nop4 into two fragments. One fragment contained RRM1 and RRM2 (Nop4 RRM 1–2), and the second fragment

contained RRM3 and RRM4 (Nop4 RRM 3–4; *Figure 4A*). We also attempted to determine if RRM3 alone was sufficient to mediate protein-protein interactions, however, RRM3 can not be stably expressed from the yeast two-hybrid vector (data not shown).

We found that Nop4 RRM 3–4 mediated the protein-protein interactions observed in the LSU processome interactome using a directed Y2H assay. Full-length Nop4 WT, Nop4 RRM 1–2 and Nop4 RRM 3–4 were expressed as prey fusion proteins (*Figure 4B*) and tested with 23 Nop4 interacting proteins (*Table 1*; *McCann et al., 2015*) expressed as bait fusions (*James et al., 1996*). Each prey was mated against each bait in an array-based Y2H assay and interactions were identified by growth on selective medium (*Figure 4C*; *de Folter and Immink, 2011*; *McCann et al., 2015*). As was previously observed in the LSU processome interactome, Nop4 WT interacted with the majority of the defined Nop4 interacting proteins after two weeks (*Figure 4C*; *McCann et al., 2015*). To our surprise, Nop4 RRM 1–2 did not interact with any of the Nop4 interacting proteins. In contrast, Nop4 RRM 3–4 interacted with the majority of the defined interacting set of proteins, similar to the

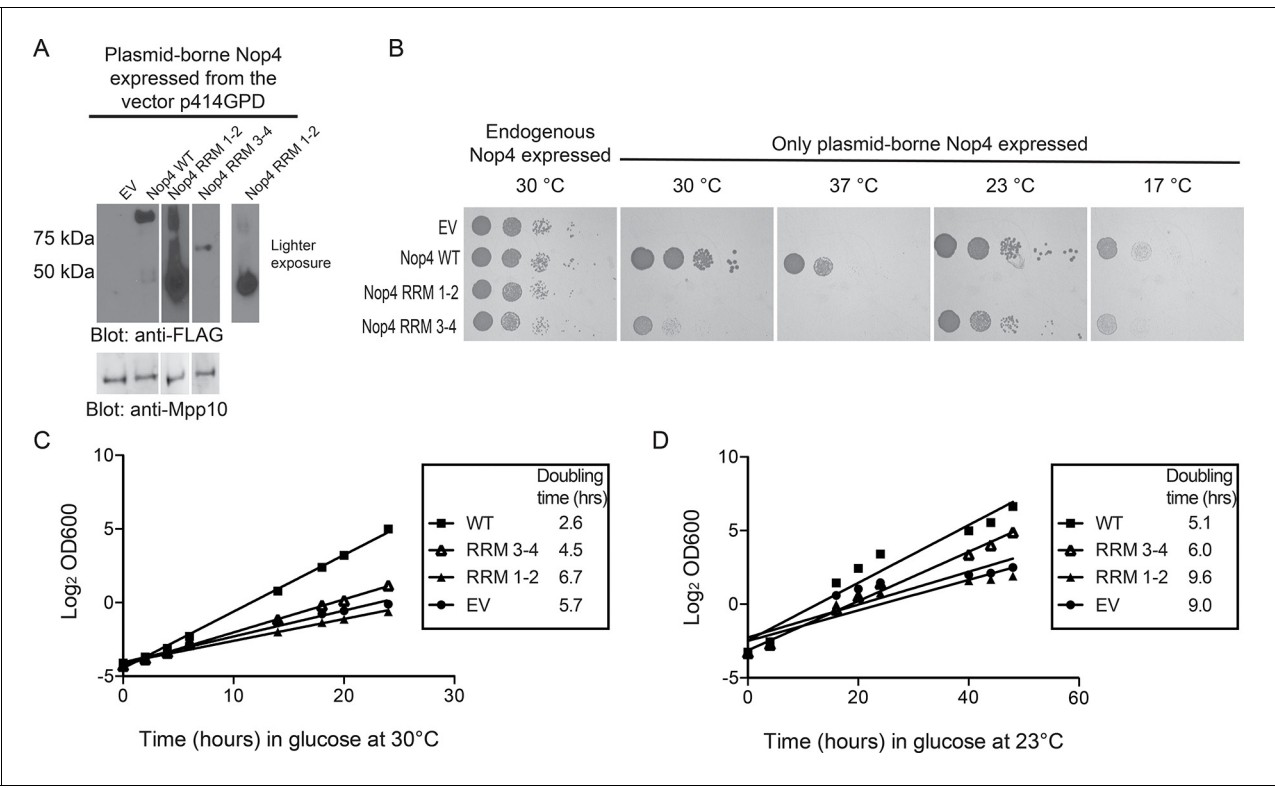

**Figure 5.** RRM3 and RRM4 of Nop4 are necessary and sufficient to complement the growth defect due to Nop4 depletion. (**A**) Nop4 WT and the Nop4 fragments are differentially expressed from the yeast expression vector p414GPD-3xFLAG-GW. Total protein was extracted from YPH499 *GAL::3xHA-NOP4* yeast transformed with empty vector (EV) or expressing Nop4 WT (78 kDa), Nop4 RRM 1–2 (28.3 kDa) or Nop4 RRM 3–4 (49.4 kDa) from the yeast expression vector, p414GPD-3xFLAG-GW. Protein extracts were separated by SDS-PAGE and analyzed by α-FLAG western blot. As a loading control, a western blot using α-Mpp10 was performed. The expression levels of Nop4 WT, Nop4 RRM 1–2 and Nop4 RRM 3–4 relative to Mpp10 were quantitated and normalized to Nop4 WT: Nop4 WT = 1, Nop4 RRM 1–2 = 4.9, Nop4 RRM 3–4 = 0.27. (**B**) Serial dilutions of yeast expressing the indicated Nop4 fragments were grown on solid medium for 3 days at 30°C and 37°C or for 5 days at 23°C and 17°C. (**C**) Yeast expressing the indicated Nop4 fragments were transferred from SG/R-Trp-Leu to SD-Trp-Leu to deplete the endogenous Nop4. Growth was monitored for 24 hr at 30°C by measuring the absorbance at $OD_{600}$. The $\log_2$ of the $OD_{600}$ was plotted over time and the slope was used to estimate the doubling time. Three biological replicates were performed starting with transformation of the plasmids into the yeast strain. (**D**) Yeast expressing the indicated Nop4 fragments were transferred from SG/R-Trp-Leu to SD-Trp-Leu to deplete the endogenous Nop4. Growth was monitored for 48 hr at 23°C by measuring the absorbance at $OD_{600}$. The $\log_2$ of the $OD_{600}$ was plotted over time and the slope was used to estimate the doubling time. Three biological replicates were performed starting with transformation of the plasmids into the yeast strain.

The following figure supplement is available for figure 5:

**Figure supplement 1.** Nop4 RRM 1–2 fails to complement even when targeted to the nucleus.

full-length Nop4 WT (*Figure 4C*). Thus, Nop4 RRMs 3 and 4 are necessary and sufficient for these protein-protein interactions and are thereby likely to mediate Nop4's function as a hub protein in the LSU processome.

## RRM3 and RRM4 are sufficient for Nop4's essential function

Nop4 RRM 3–4 is also sufficient to complement the growth defect observed upon Nop4 depletion. We constitutively expressed either full length Nop4 WT, Nop4 RRM 1–2 or Nop4 RRM 3–4 from plasmids in a yeast strain in which endogenous Nop4 was depleted (*Figure 1B*). Western blotting of total protein demonstrated that plasmid-borne, FLAG-tagged Nop4 WT, Nop4 RRM 1–2 and Nop4 RRM 3–4 were expressed, albeit at very different levels (*Figure 5A*). Serial dilutions of strains bearing the plasmids: empty vector (EV), Nop4 WT, Nop4 RRM 1–2 and Nop4 RRM 3–4 were spotted onto plates containing glucose and incubated at 30°C, 37°C, 23°C and 17°C. As expected, EV did not complement the growth defect at any temperature whereas Nop4 WT complemented at all temperatures (*Figure 5B*). Like EV, Nop4 RRM 1–2 did not complement the growth defect. However, Nop4 RRM 3–4 complemented the growth defect at 23°C and 17°C and partially complemented at 30°C (*Figure 5B*).

As an additional test, we analyzed complementation in liquid medium at 30°C and 23°C. Endogenous Nop4 was depleted and the growth of strains expressing no Nop4 (empty vector; EV), Nop4 WT, Nop4 RRM 1–2 and Nop4 RRM 3–4, was monitored for 24 hr at 30°C or 48 hr at 23°C (*Figure 5C,D*). Similar to results on solid medium, Nop4 RRM 3–4 did not significantly complement the growth defect at 30°C but did complement the growth defect at 23°C, whereas Nop4 RRM 1–2 did not complement at either temperature (*Figure 5C,D*). The failure of Nop4 RRM 1–2 to complement the growth defect is not due to aberrant localization of the protein fragment. Nop4 RRM 1–2 also fails to complement when expressed from the yeast two-hybrid vector, pACT2, which ensures targeting of the fragment to the nucleus (*Figure 5—figure supplement 1A,B*), suggesting that this domain is not essential for Nop4 function. In contrast, the ability of Nop4 RRM 3–4 to complement the growth defect in both solid and liquid medium suggests that the essential function of Nop4 is mediated through RRMs 3 and 4.

Expression of Nop4 RRM 3–4 is sufficient to partially restore pre-rRNA processing. To determine whether complementation of the growth defect is due to rescue of the pre-rRNA processing defect, total RNA was harvested from strains depleted of endogenous Nop4 for 48 hr at 23°C or for 24 hr at 30°C and bearing plasmids expressing no Nop4 (EV), Nop4 WT, Nop4 RRM 1–2 or Nop4 RRM 3–4. The 25S and 18S rRNAs were visualized by ethidium bromide staining, quantified and the ratio of 25S/18S was calculated and normalized to WT for each time point (*Figure 6A,B*). Complementation of growth correlated with the rescue of pre-rRNA processing. As expected, Nop4 WT restored the 25S/18S ratio compared to EV at both 23°C and 30°C. Nop4 RRM 1–2 did not complement growth and did not rescue the 25S/18S ratio at either temperature (*Figure 6A,B*). In contrast, Nop4 RRM 3–4 was sufficient to significantly rescue the 25S/18S ratio compared to EV at 23°C, but not at 30°C (*Figure 6B*), consistent with a restoration of 25S levels at 23°C.

To further assess the rescue of pre-rRNA processing, we also examined pre-rRNA processing by northern blotting using an oligonucleotide probe in ITS2 (*Figure 2A*). Depletion of Nop4 (EV) led to a reduction in the 27S and 7S pre-rRNAs but did not affect the levels of the 35S pre-rRNA, as has been observed before (*Bergès et al., 1994*; *Sun and Woolford, 1994*; *Qiu et al., 2008*). Expression of Nop4 WT restored pre-rRNA processing and the levels of the 27S and 7S pre-rRNA intermediates compared to the EV control at both 23°C and 30°C (*Figure 6A,B*). Strikingly, expression of Nop4 RRM 3–4, but not Nop4 RRM 1–2, significantly restored the levels of the 27S and 7S intermediates at 23°C but failed to rescue at 30°C (*Figure 6B*). Thus, pre-rRNA processing parallels the observed growth complementation and suggests that the essential function of Nop4 in ribosome assembly is mediated through the protein binding RRMs, RRM3 and RRM4.

## The ANE syndrome mutation disrupts the structure of human RBM28 RRM3

As the ANE syndrome mutation in yeast Nop4 causes pre-rRNA processing defects and reduces protein-protein interactions with a subset of proteins that we tested, we hypothesized that the mutation causes a structural change in RRM3. To test this, we analyzed WT and mutant human RBM28 RRM3

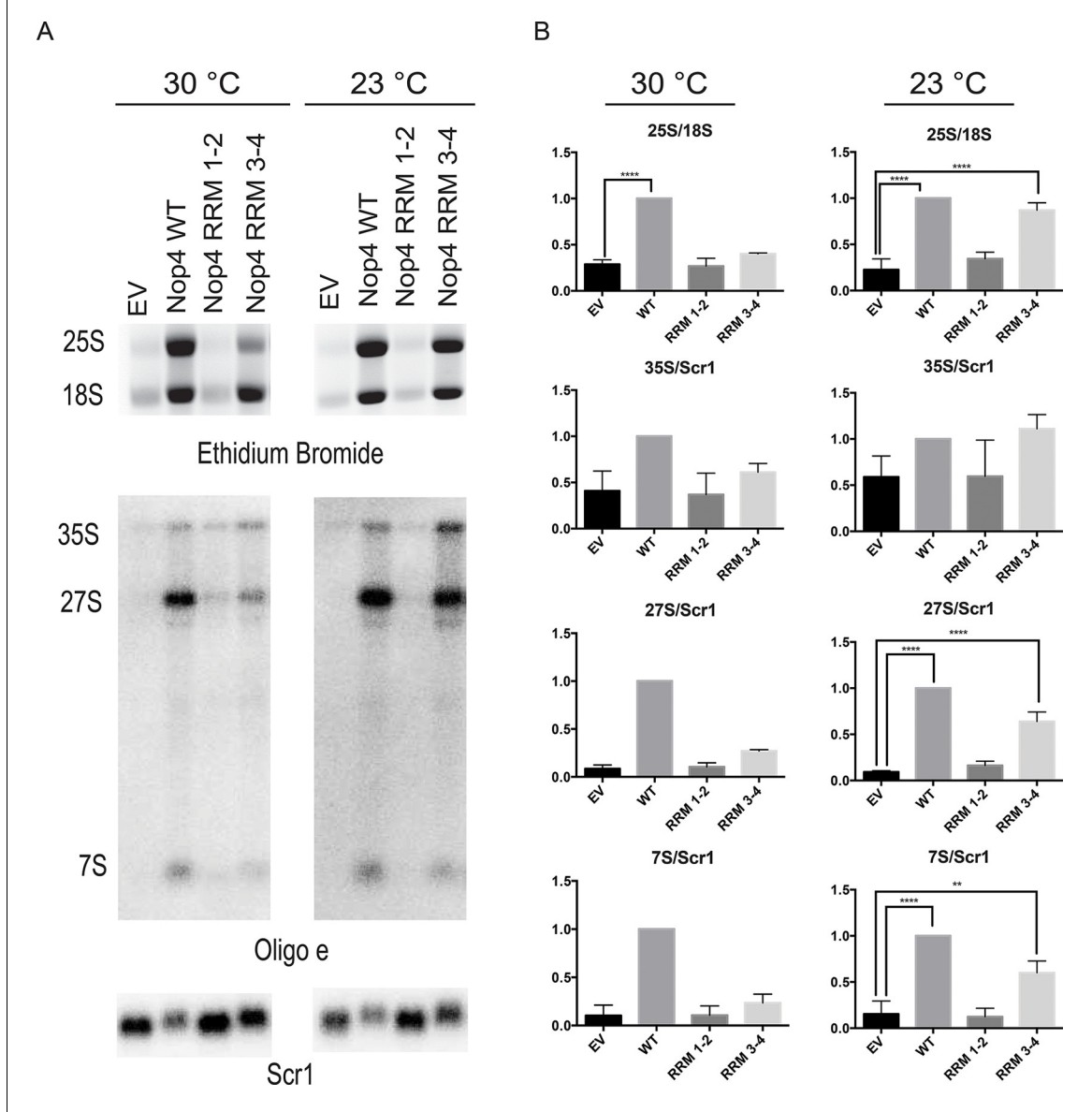

**Figure 6.** Nop4 RRM 3–4 is necessary and sufficient to complement the pre-rRNA processing defect after Nop4 depletion. (**A**) Top panel: Total RNA extracted from yeast expressing the indicated Nop4 fragment after depletion of endogenous Nop4 for 24 hr at 30°C or for 48 hr at 23°C was visualized by ethidium bromide staining. Bottom panel: Northern blot analysis of total RNA using oligonucleotide probe e, which is complementary to a region of ITS2 of the pre-rRNA. As a loading control, we used an oligonucleotide complementary to the Scr1 RNA. Three biological replicates were performed. (**B**) The ratios of the mature rRNAs (25S/18S) and the ratios of the precursors to the loading control Scr1 (35S/Scr1, 27S/Scr1S and 7S/Scr1) were calculated from three replicate experiments and were plotted with error bars representing the standard deviation. The significance of the ratios of 25S/18S, 35S/Scr1, 27S/Scr1 and 7S/Scr1 of Nop4 WT, Nop4 RRM 1–2 or Nop4 RRM 3–4 compared to EV was evaluated using one-way ANOVA. ****indicates a p value < 0.0001. **indicates a p value < 0.01.

The following source data is available for figure 6:

**Source data 1.** Quantitation and statistical analyses for *Figure 6B*.

domains (amino acids 330–419) by circular dichroism (CD) and NMR. We used RBM28 RRM3 because Nop4 RRM3 was not soluble. We found that the L351P mutation disrupts the backbone structure of the RBM28 RRM3 domain. The CD spectra of WT and L351P RBM28 RRM3 showed notable differences in ellipticities [θ] at 208, 215 and 222 nm (*Figure 7A*), indicating that the amino

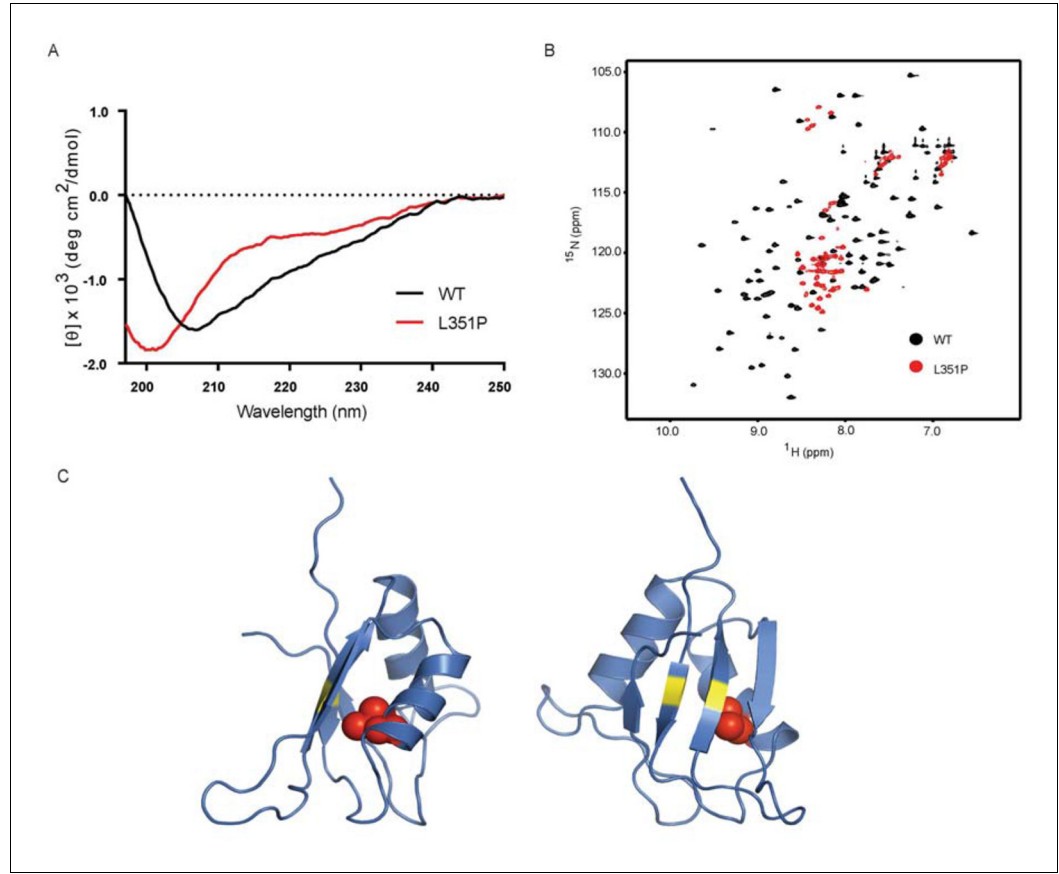

**Figure 7.** The ANE syndrome mutation, L351P, in human RBM28 disrupts RRM3 domain structure. (**A**) Circular dichroism spectra of WT human RBM28 RRM3 (black) and L351P mutant (red). Four technical replicates were performed. (**B**) $^{15}$N-HSQC spectra of WT hRBM28 RRM3 (amino acids 330–419) (black) and L351P mutant (red) are superimposed and plotted at the same contour level. In addition to clustering of resonances around 8.0~8.5 ppm in the proton dimension, dispersion of glutamine and asparagine side chains (7.0~7.8 ppm in the $^{1}$H dimension and 111~114 ppm in the $^{15}$N dimension) is reduced considerably, consistent with protein backbone disruption. Thirty-two technical replicates were performed. (**C**) Ribbon diagram of a homology model of human RBM28 RRM3. The model including residues 330–419 was generated using the Phyre2 server (*Kelley et al., 2015*), and the best template was a solution structure of mouse RBM28 RRM3 (90% sequence identity with human RBM28, PDB ID 1X4H). L351 is shown with red space-filling spheres and typical RNA interacting residues in RNP motifs are colored yellow.

acid substitution reduced α-helical and β-sheet content. The CD spectrum of the L351P mutant protein with a minimum only at ~200 nm indicated the presence of random coil. We confirmed the disruption of domain folding by NMR. $^{15}$N-HSQC of WT RBM28 RRM3 showed well dispersed resonances (*Figure 7B*), indicating the presence of both α-helices and β-strands, as expected for an RRM domain(*Nagai et al., 1990*). In contrast, the majority of resonances in the spectrum of L351P ANE syndrome mutant RRM3 were clustered around 8.0~8.5 ppm in the proton dimension, demonstrating that the mutant protein backbone is disordered. A homology model of the human RBM28 RRM3, based on an NMR structure of mouse RBM28 RRM3, indicates that L351 is buried within the core of the domain (*Figure 7C*). Mutation to proline would be expected to disrupt helix α1 and the overall tertiary structure of this RRM.

## Discussion

We determined that the molecular pathogenesis of ANE syndrome (L351P in the nucleolar protein, RBM28) results from disrupted RRM3 protein structure that reduces its function as a hub protein in

the LSU processome and causes defects in pre-rRNA processing. Introducing the ANE syndrome mutation into RRM3 of the yeast ortholog, Nop4 (L306P), causes growth and pre-rRNA processing defects, as well as reduced association with interacting protein partners. RRMs 3 and 4 of Nop4 alone are necessary and sufficient for Nop4 protein-protein interactions and for yeast growth and pre-rRNA processing. Biophysical methods indicate that RRM3 containing the ANE syndrome mutation is unfolded. Taken together, these results provide evidence that ANE syndrome is a ribosomopathy with a molecular defect in nucleolar steps of ribosome biogenesis.

The finding that an essential function of Nop4 is to mediate protein binding was unexpected. Nop4 has been cross-linked to pre-rRNA (*Bergès et al., 1994*; *Sun and Woolford, 1994*; *Granneman et al., 2011*), leading to the expectation that its 4 RRMs would be essential for RNA binding. We show by Y2H analysis that the C-terminal half of Nop4 (RRMs 3 and 4) mediates protein-protein interactions and hub protein function in the LSU processome. Strikingly, this half of Nop4 is necessary and sufficient to complement the growth and pre-rRNA processing defects observed upon depletion of endogenous Nop4. As the ANE syndrome mutation occurs in RRM3 and disrupts Nop4 protein-protein interactions, ANE syndrome is therefore likely a disease of altered protein interaction rather than of RNA binding. We are now in a position to determine which interactions are critical for Nop4 function and how disruptions of those specific interactions contribute to ANE syndrome pathogenesis.

Nevertheless, Nop4 is undoubtedly an RNA-binding protein. Nop4 binds RNA in vitro (*Sun and Woolford, 1997*), co-immunoprecipitates the 27S and 7S pre-rRNAs and crosslinks to the 25S within the pre-rRNA in vivo (*Granneman et al., 2011*). Additionally, all 4 RRMs are important for Nop4 function as mutations in any of the RRMs disrupt growth and LSU assembly at 37°C (*Sun and Woolford, 1997*). Although RRMs 1 and 2 are not required for Nop4's hub protein function or sufficient for growth, they may bind the pre-rRNA. Furthermore, while RRMs 3 and 4 mediate protein binding, the possibility that they may also bind RNA is not precluded.

How does an RRM facilitate protein binding and thus hub protein function? Several examples of RRMs mediating interactions with other proteins have been identified including in the U2AF[35]-U2AF[65], the U2AF[65]-SF1, the Snu17-Bud13, and the Y14-Mago complexes (*Kielkopf et al., 2001*; *Fribourg et al., 2003*; *Lau et al., 2003*; *Selenko et al., 2003*; *Tripsianes et al., 2014*). An RRM is comprised of a 4-stranded β-sheet, which forms the primary RNA binding interface, and 2 α-helices (*Maris et al., 2005*; *Cléry et al., 2008*). The α-helices of the RRM often mediate interaction between protein pairs, leaving the β-sheet accessible for RNA binding (*Kielkopf et al., 2001*; *Selenko et al., 2003*; *Tripsianes et al., 2014*). Alternatively, in the case of the Y14-Mago complex, the interaction is through the β-sheet, which precludes RNA binding (*Fribourg et al., 2003*; *Lau et al., 2003*). RRMs also can serve as oligomerization domains. In the case of Human antigen R (HuR), the third RRM promotes dimerization through its α-helices (*Scheiba et al., 2014*). These examples highlight the possibility that RRMs 3 and 4 of Nop4 may mediate protein binding and hub function through more than one of its structural motifs. The ANE mutation lies in an α-helix and the mutation disrupts not only the α-helix, but unfolds the RRM tertiary structure. Since the mutation abrogates some but not all protein-protein interactions, a subset of Nop4 interactions may be mediated by unstructured peptide elements in RRM3 or by the RRM4 domain.

Fibroblasts from patients with ANE syndrome have ribosome levels reduced to approximately 60% of controls (*Nousbeck et al., 2008*), consistent with the modest defects in ribosome biogenesis that we can now associate with the ANE syndrome mutation. In the yeast model system, the mutation confers reduced growth and mild pre-rRNA processing defects when compared to the Nop4 null (EV). Thus, the ANE syndrome mutation is a hypomorphic allele, consistent with its autosomal recessive inheritance (*Nousbeck et al., 2008*). This hypomorphism may, in part, explain how the ANE syndrome mutation is compatible with life, as the presence of the mutation leads to a partially functional RBM28 protein.

## Materials and methods

### Amino acid sequence alignment

The amino acid sequence of Nop4 was obtained from the Saccharomyces Genome Database (www.yeastgenome.org). The amino acid sequence of human RBM28 (accession number NP_060547) was

obtained from the Protein Database (http://www.ncbi.nlm.nih.gov/protein/). The amino acid sequences of RBM28 from M. mulatta, M. musculus, X. tropicalis and D. rerio were obtained from Uniprot (www.uniprot.org). Amino acid alignments were determined using either ClustalX (*Jeanmougin et al., 1998*) or MegAlign Pro version 12.2.0 from DNASTAR. Madison, WI.

## Yeast strains and plasmids

A *GAL::3HA-NOP4* strain was generated in the parental strain YPH499 (*MATa ura3-52 lys2-801 ade2-101 trp1-Δ63 his3-Δ200 leu2-Δ1*) as described in (*Charette and Baserga, 2010*) that expresses 3HA-tagged Nop4 from the endogenous locus when grown in medium containing galactose but represses endogenous Nop4 expression when grown in medium containing glucose. The strain was confirmed by western blot using α-HA-HRP (Roche, Indianapolis, Indiana ).

*NOP4* was shuttled into the Gateway-modified yeast expression vector p414GPD-3xFLAG-GW (TRP1) or the Gateway-modified Y2H prey vector pACT2 (LEU2) and RBM28 was shuttled into p414GPD-3xFLAG-GW (TRP1) by Gateway cloning (Life Technologies) as in (*Charette and Baserga, 2010*). Site-directed mutagenesis to introduce the L306P missense mutation was performed using a Change-IT kit (Affymetrix, Santa Clara, California). RBM28, Nop4 WT and Nop4 L306P were all fully sequenced by either the W.M. Keck Foundation facility at the Yale School of Medicine or by GENEWIZ, Inc. Expression of RBM28, Nop4 WT and Nop4 L306P from p414GPD-3xFLAG-GW or pACT2 was analyzed by western blot using either α-3xFLAG-HRP (Sigma, St. Louis, Missouri) or α-HA-HRP (Roche, Indianapolis, Indiana). As a loading control, a western blot using α-Mpp10 (*Dunbar et al., 1997*) was performed.

The Nop4 fragments in *Figure 4A* were cloned into a Gateway Entry vector (pDONR221) and subsequently shuttled into the Y2H prey vector (pACT2) or the yeast expression vector p414GPD-3xFLAG-GW by Gateway cloning (Life Technologies) as in (*Charette and Baserga, 2010*). All clones were fully sequenced by either the W.M. Keck Foundation facility at the Yale School of Medicine or by GENEWIZ, Inc. Expression of the Nop4 fragments from p414GPD-3xFLAG-GW and pACT2 was analyzed by western blot using either α-3xFLAG-HRP (Sigma, St. Louis, Missouri) or α-HA-HRP (Roche). As a loading control, a western blot using α-G-6-PDH (Sigma) or using α-Mpp10 (*Dunbar et al., 1997*) was performed.

## Growth assays

For analysis of the complementation by RBM28 of the growth defect conferred by Nop4 depletion, the YPH499 *GAL::3xHA-NOP4* yeast strain was transformed with either empty vector (EV) or plasmids expressing Nop4 or RBM28. For serial dilutions, 0.2 mL of cells at an $OD_{600}$ of 1 were resuspended in 1 mL water, diluted 1/10 and spotted onto SG/R-Trp or SD-Trp. Cells were incubated at 30°C or 37°C for 3 days or at 23°C or 17°C for 5 days. Two biological replicates were performed starting with transformation of the plasmids into the yeast strain.

For analysis of the effect of the ANE syndrome mutation (L306P) on growth, the YPH499 *GAL::3xHA-Nop4* yeast strain was transformed with either empty vector (EV), or plasmids expressing Nop4 WT or Nop4 L306P. For serial dilutions, 0.2 mL of cells at an $OD_{600}$ of 1 were resuspended in 1 mL water, diluted 1/10 and spotted onto medium containing 2% w/v galactose and 2% w/v raffinose and lacking tryptophan (SG/R-Trp) or onto medium containing 2% w/v glucose (dextrose) and lacking tryptophan (SD-Trp). Cells were incubated at 30°C or 37°C for 3 days or at 23°C or 17°C for 5 days. Four biological replicates were performed starting with transformation of the plasmids into the yeast strain. For analysis in liquid medium, the *GAL::3xHA-NOP4* yeast strain transformed with empty vector (EV) or expressing either 3xFLAG-tagged Nop4 WT or Nop4 L306P from the p414GPD vector was depleted of endogenous Nop4 by first growing cultures to mid-log phase ($OD_{600}$ = 0.4–0.8) in SG/R-Trp at 30°C and then transferring the cultures to the non-permissive (SD-Trp) medium and 23°C. The cells were maintained in mid-log phase ($OD_{600}$ < 0.8) by dilution of the culture with fresh SD-Trp media. Growth was monitored by $OD_{600}$ measurement for 48 hr. Three biological replicates were performed starting with transformation of the plasmids into the yeast strain.

For analysis of the complementation by the Nop4 fragments of the growth defect conferred by Nop4 depletion, the YPH499 *GAL::3xHA-NOP4* yeast strain was co-transformed with empty p415GPD-3xHA-GW and either empty p414GPD-3xFLAG-GW vector (EV) or p414GPD-3xFLAG-GW

expressing the Nop4 fragments. For serial dilutions, 0.2 mL of cells at an $OD_{600}$ of 1 were resuspended in 1 mL water, diluted 1/10 and spotted onto SG/R-Trp-Leu or SD-Trp-Leu. Cells were incubated at 30°C or 37°C for 3 days or at 23°C or 17°C for 5 days. Two biological replicates were performed starting with transformation of the plasmids into the yeast strain. For analysis in liquid medium of the complementation by the Nop4 fragments of the growth defect conferred by Nop4 depletion, the YPH499 *GAL::3xHA-NOP4* yeast strain expressing one of the 3xFLAG-tagged Nop4 fragments was depleted of endogenous Nop4 by first growing cultures to mid-log phase ($OD_{600}$ = 0.4–0.8) in SG/R-Trp-Leu at 30°C and then transferring the cultures to the non-permissive (SD-Trp-Leu) medium and growing at either 30°C or 23°C. The cells were maintained in mid-log phase ($OD_{600}$ < 0.8) by dilution of the culture with fresh SD-Trp-Leu media. Growth was monitored by $OD_{600}$ measurement for 24 hr at 30°C or for 48 hr at 23°C. Three biological replicates were performed starting with transformation of the plasmids into the yeast strain.

For analysis of the complementation of the growth defect conferred by Nop4 depletion by the Nop4 fragments expressed from pACT2, the YPH499 *GAL::3xHA-NOP4* yeast strain was transformed with either pACT2 Nop4 WT, pACT2 Nop4 RRM 1–2 or pACT2 Nop4 RRM 3–4. For serial dilutions, 0.2 mL of cells at an $OD_{600}$ of 1 were resuspended in 1 mL water, diluted 1/10 and spotted onto SG/R- Leu or SD- Leu. Cells were incubated at 30°C or 37°C for 3 days or at 23°C or 17°C for 5 days. Two biological replicates were performed starting with transformation of the plasmids into the yeast strain.

## RNA and northern blot analysis

For analysis of the effect of the ANE syndrome mutation, the YPH499 *GAL::3xHA-NOP4* yeast strain was transformed with EV or plasmids expressing either Nop4 WT or Nop4 L306P from the p414GPD-3xFLAG vector. The strain was depleted of endogenous Nop4 as described above. Cells (20 mL) at an $OD_{600}$ of ∼0.5 were collected from each culture after 0 and 48 hr of growth at 23°C. Total RNA was extracted as described in (*Dunbar et al., 1997*). For analysis of the mature rRNAs, 5 µg of total RNA per sample was separated by electrophoresis on a 1% agarose gel. RNA was visualized by ethidium bromide staining, and the bands were quantified by densitometric analysis using ImageJ (*Schneider et al., 2012*). For northern blot analysis, 3 µg of total RNA per sample was separated by electrophoresis on a 1% agarose/1.25% formaldehyde gel, transferred to a nylon membrane (Hybond-XL, GE Healthcare, Buckinghamshire, England) and detected by hybridization with radiolabelled oligonucleotide e (5′ – GGCCAGCAATTTCAAGT – 3′) and radiolabelled oligonucleotide Scr1 (5′ – CGTGTCTAGCCGCGAGGAAGGATTTGTTCC – 3′) as described in (*Wehner and Baserga, 2002*; *Qiu et al., 2014*). The 35S, 27S and 7S pre-rRNAs and the Scr1 RNA were quantified using a Biorad Personal Molecular Imager. The ratios of 27S or 7S to the 35S pre-rRNA and the ratios of the 35S, 27S or 7S to Scr1 were calculated. Four biological replicates were performed for each experiment. GraphPad PRISM was used to calculate the means of the ratios and plotted with error bars (SD). Significance compared to the WT control was determined using one-way ANOVA.

For analysis of the complementation by the Nop4 fragments of the pre-rRNA processing defect, the YPH499 *GAL::3xHA-NOP4* yeast strain transformed with EV or expressing one of the 3xFLAG-tagged Nop4 fragments was depleted of endogenous Nop4 as described. Cells (20 mL) at an $OD_{600}$ of ∼0.5 were collected from each culture after either 24 hr of growth at 30°C or after 48 hr of growth at 23°C. Total RNA was extracted as described in (*Dunbar et al., 1997*). For northern blot analysis, 3 µg of total RNA per sample was separated by electrophoresis on a 1% agarose/1.25% formaldehyde gel, transferred to a nylon membrane (Hybond-XL, GE Healthcare, Buckinghamshire, England) and detected by hybridization with radiolabelled oligonucleotide probe e (5′ – GGCCAG-CAATTTCAAGT – 3′), which is complementary to ITS2 of the yeast pre-rRNA, and probe Scr1 (5′-CG TGTCTAGCCGCGAGGAAGGATTTGTTCC-3′), which is complementary to the RNA Scr1, as described in (*Wehner and Baserga, 2002*). The 7S, 27S and 35S pre-rRNA species were quantified on a Biorad Personal Molecular Imager, and the ratios of 7S, 27S or 35S to Scr1 were calculated. Three biological replicates were performed for each experiment. GraphPad PRISM was used to calculate the means of the ratios and plotted with error bars (SD). Significance compared to the EV control was determined using one-way ANOVA.

## Yeast two-hybrid analysis

For Y2H analysis to determine the effect of the ANE syndrome mutation (L306P; *Figure 3*) on protein-protein interactions, a subset of the Nop4 interacting proteins identified in (*McCann et al., 2015*) were expressed from the Y2H bait vector (pAS2-1) and were co-transformed with either empty pACT2 vector, Nop4 WT or Nop4 L306P into the yeast strain PJ69-4α. Co-transformed yeast were serially diluted by resuspending 0.2 mL of cells at an $OD_{600}$ of 1 in 1 mL water, diluting 1/10 and spotting onto medium selecting for the presence of both plasmids (SD-Leu-Trp) and medium selecting for Y2H interactions [SD-Leu-Trp-His + 6 mM 3-Amino-1,2,4 triazole (3-AT)]. Cells were incubated at 30°C for 7 days. Two biological replicates of a subset of interacting proteins were performed starting with co-transformation of the bait and prey plasmids into the Y2H strain.

For Y2H analysis of Nop4 fragments (*Figure 4*), Nop4 WT and the Nop4 fragments were shuttled into the Y2H prey vector (pACT2) by Gateway (Invitrogen) recombination and individually transformed into the yeast strain PJ69-4a. The Nop4 interacting proteins identified in (*McCann et al., 2015*) were shuttled into the Y2H bait vector (pAS2-1) and transformed into the yeast strain PJ69-4α as an array. All baits were mated against all preys in a semi-high-throughput Y2H matrix screen (*de Folter and Immink, 2011*). The mated yeast were transferred to SD-Leu-Trp plates to select for diploids bearing both the bait and prey vectors. Diploids were then transferred to the selective medium: SD-Leu-Trp-His + 6mM 3-AT. Growth on selective medium greater than that of the negative control after 2 weeks was indicative of an interacting bait-prey pair. Two biological replicates were performed starting with the transformation of the bait and prey plasmids into the Y2H strains.

## Co-immunoprecipitations

A subset of the Nop4 interacting proteins identified in (*McCann et al., 2015*) were expressed from p414GPD-3xFLAG-GW and were co-transformed with either p415GPD-3xHA-GW Nop4 WT or Nop4 L306P into the yeast strain YPH499 (*MATa ura3-52 lys2-801 ade2-101 trp1-Δ63 his3-Δ200 leu2-Δ1*). The resulting transformed strains were grown in medium containing 2% dextrose and lacking leucine and tryptophan (SD-Leu-Trp) at 30°C. Negative control strains were only transformed with p415GPD-3xHA-GW clones and were grown in medium containing 2% dextrose and lacking leucine (SD-Leu) at 30°C. For each co-immunoprecipitation, 20 mL of cells at an $OD_{600}$ of ~0.5 was collected, washed with water and resuspended in NET2 (20 mM Tris-HCl, pH 7.5, 150 mM NaCl, 0.01% Nonidet P-40) with 1x HALT protease inhibitors (Thermo Fisher Scientific, Rockford, Illinois). Cells were lysed with 0.5-mm glass beads. The lysate was cleared by centrifugation at 15,000g for 10 min at 4°C. Aliquots of 500 μL of lysate were incubated with α-FLAG beads (Sigma) for 1 hr at 4°C. The beads were washed five times with NET2 and resuspended in 25 μL SDS loading dye. Immunoprecipitates were separated on 4–12% Bis-Tris PAGE and transferred to a PVDF membrane. Western blot analysis with α-HA (Abcam, Cambridge, Massachusetts) and α-FLAG-HRP (Sigma) was performed. The protein bands were quantified using ImageJ (*Schneider et al., 2012*) and the ratio of HA to FLAG was calculated. GraphPad PRISM was used to plot the means of the ratios with error bars (SD). Significance compared to the WT control for each interacting protein was determined using a t-test. Three biological replicates were performed.

## Immunofluorescence

For analysis of the Nop4 fragment localization, the YPH499 *GAL::3xHA-NOP4* yeast strain was transformed with either pACT2 Nop4 WT, pACT2 Nop4 RRM 1–2 or pACT2 Nop4 RRM 3–4 and endogenous Nop4 was depleted for 48 hr at 23°C as described above. For immunofluorescence, 50 mL of cells at an $OD_{600}$ of ~0.5 was collected, washed with water, resuspended in Fixing buffer (100 mM Sucrose, 5% paraformaldehyde) and incubated at room temperature for 45 min. The cells were then washed three times with Buffer B (100 mM $K_2HPO_4$ pH 7.5, 1.2 M sorbitol), resuspended in 1 ml of Spheroplasting buffer (100 mM $K_2HPO_4$ pH 7.5, 1.2 M sorbitol, 30 mM β-mercaptoethanol) containing lyticase (Sigma) at 800 U/mL and incubated for 8 min at 30°C. The reaction was stopped by adding 5 mL of ice-cold Buffer B. The yeast were washed once with ice-cold Buffer B, resuspended in 1 ml of Buffer B and 500 μL were plated into each well of a 12-well plate containing a poly-D-lysine coated cover glass. Cells were incubated for 1 hr at 4°C, washed once with Buffer B and permeabilized overnight in 70% ethanol at -20°C. Fixed and permeabilized cells incubated with mouse anti-HA.11 (BioLegend, San Diego, California) diluted 1:1000 for 90 min at room temperature. The

secondary antibody (Alexa Fluor 488 donkey anti-Mouse; Life Technologies, Carlsbad, California) was used at a dilution of 1:1000 and was incubated for 1 hr at room temperature. Cover glasses were mounted with Prolong Gold containing DAPI (Life Technologies) and cells were imaged on a wide-field, epifluorescence microscope using a x100 oil-immersion objective (Carl Zeiss).

## Protein expression and purification

*E. coli* codon-optimized cDNAs encoding the wild type (WT) and L351P mutant human RBM28 RRM3 (330-419) were obtained by gene synthesis (Genewiz, Inc.). The cDNAs were subcloned into pSMT3 with an N-terminal $His_6$-SUMO tag. WT and L351P human RBM28 RRM3 domains were over-expressed in *E. coli* strain BL21-CodonPlus (DE3)-RIL (Agilent Technologies, Santa Clara, California) at 20 °C overnight after induction with 0.5 mM IPTG. The cells were collected by centrifugation, and pellets were resuspended in lysis buffer (50 mM Tris-HCl, pH 8.0, 500 mM NaCl) and stored at −80 °C until use.

Cells expressing WT human RBM28 RRM3 domain were disrupted by sonication. The soluble fraction was applied to a Ni-NTA agarose column and thoroughly washed with lysis buffer containing 20 mM imidazole. The target SUMO fusion protein was eluted with lysis buffer containing 400 mM imidazole. The fusion protein was cleaved overnight with 0.2 mg of Ulp1 protease. The cleaved fusion protein sample was applied to a HiLoad 16/60 Superdex 75 column (GE Healthcare) equilibrated with lysis buffer containing 1 mM Tris (2-carboxyethyl) phosphine hydrochloride (TCEP). The eluted fractions containing WT hRBM28 RRM3 protein were pooled and applied to a Ni-NTA agarose column again to remove released SUMO protein. The protein sample was dialyzed against a buffer containing 50 mM Tris-HCl, pH 8.0, 100 mM NaCl and 0.5 mM TCEP and purified further using a HiTrap Q HR anion-exchange column (GE Healthcare). Bound proteins were eluted using a linear gradient from 0.05 to 1 M NaCl in 50 mM Tris-HCl, pH 8.0 and 0.5 mM TCEP. Peak fractions containing WT hRBM28 RRM3 were pooled and concentrated.

L351P mutant human RBM28 RRM3 domain was purified by the same procedure as WT protein up to the first Ni-NTA agarose column. The eluted SUMO fusion protein was cleaved overnight with Ulp1 protease in conjunction with dialysis into 50 mM Tris-HCl, pH 8.0, 100 mM NaCl and 0.5 mM TCEP. The cleaved fusion protein sample was applied to a HiTrap Q HR anion-exchange column and eluted with a linear gradient from 0.1 to 1 M NaCl in 50 mM Tris-HCl, pH 8.0 and 0.5 mM TCEP. The eluted fractions containing L351P hRBM28 RRM3 protein were pooled and reapplied to a Ni-NTA agarose column. The protein was concentrated and purified further using a HiLoad 16/60 Superdex 75 column equilibrated with lysis buffer containing 0.5 mM TCEP. Two peaks of L351P hRBM28 RRM3 eluted from the Superdex 75 column. Because the peak eluting at 58.5 ml contained many contaminating proteins, only the fractions containing the peak eluting at 72.2 ml were pooled and concentrated.

## Circular dichroism (CD) spectroscopy

The CD spectra of WT and L351P mutant human RBM28 RRM3 domains were measured on a JASCO J-810 CD spectrometer at room temperature. For each sample (200 μL in a 0.1 cm light-path cell), four scans were accumulated in the wavelength range of 190–260 nm with a 0.2 nm step size. Protein samples were 100 μg/mL in 20 mM Na phosphate buffer, pH 7.0, 100 mM NaCl and 0.2 mM TCEP. The raw CD data were adjusted by subtracting a buffer blank. Four technical replicates were performed.

## NMR measurement

$^{15}N$-/$^{13}C$-labeled WT and L351P human RBM28 RRM3 domains were prepared as described above, except that *E. coli* cultures were grown in M9 medium containing appropriate isotopes. The protein samples for NMR experiments were 0.4 mM in 20 mM sodium phosphate buffer, pH 7.0, 100 mM NaCl, 0.2 mM TCEP and 10% (v/v) $D_2O$. $^{15}N$-HSQC spectra were collected on a Varian Inova 60 MHz magnet installed with a cryo-probe at 298 K. The data were processed and plotted using NMRPipe (*Delaglio et al., 1995*) and NMRViewJ (*Johnson, 2004*). Thirty-two technical replicates were performed.

## Acknowledgements

We thank R Petrovich of the NIEHS Protein Expression Core Facility for assistance with CD measurements, C Tucker and R Wine for assistance with immunofluorescence, E DeRose for assistance with the NMR data collection and Baserga laboratory members for critical discussion and reading the manuscript. This work was supported by National Institute of General Medical Sciences grant 0115710 (SJB), the Intramural Research Program of the National Institutes of Health, National Institute of Environmental Health Sciences (TMTH) and the National Institute of Health (NIH) predoctoral training grant in genetics T32 GM 007499 (KLM).

## Additional information

### Funding

| Funder | Grant reference number | Author |
|---|---|---|
| National Institute of Environmental Health Sciences | 1ZIAES050165 | Takamasa Teramoto<br>Jun Zhang<br>Traci M Tanaka Hall |
| National Institute of General Medical Sciences | 0115710 | Kathleen L McCann<br>Susan J Baserga |
| National Institutes of Health | T32 GM 007499 | Kathleen McCann |

The funders had no role in study design, data collection and interpretation, or the decision to submit the work for publication.

### Author contributions

KLM, TT, JZ, Conception and design, Acquisition of data, Analysis and interpretation of data, Drafting or revising the article; TMTH, SJB, Conception and design, Analysis and interpretation of data, Drafting or revising the article

### Author ORCIDs

Kathleen L McCann, http://orcid.org/0000-0002-7144-4851

## Additional files

### Supplementary files

• Supplementary file 1. Western blot quantitation source data.

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
