## [Decision Letter]

[Editors’ note: a previous version of this study was rejected after peer review, but the authors submitted for reconsideration. The previous decision letter after peer review is shown below.]

Thank you for submitting your work entitled "The molecular basis for ANE syndrome revealed by the large ribosomal subunit processome interactome" for consideration by *eLife*. Your article has been reviewed by three reviewers in addition to a Reviewing Editor, and James Manley was the Senior Editor. The decision was reached after consultation between the four reviewers. Based on these discussions, and the individual reviews below, we regret to inform you that your work cannot be considered further for publication in *eLife*.

While all reviewers thought that the subject matter was significant, there was some disagreement regarding the overall quality of the work. In particular, two of the referees felt that the experiments did not go far enough in dissecting the molecular defect caused by the mutation. They also raised a number of technical points, most importantly the question of the effect of temperature. Even the more positive reviewer raised similar technical points. The reviewers concluded that there was insufficient enthusiasm for the work to warrant publication in *eLife*. Because *eLife* does not encourage major revision we are declining the paper.

*Reviewer #1:*

In this interesting paper the authors study the biochemical consequences of a missense mutation in the nucleolar protein RBM28 known to be causative for ANE syndrome. Most of the experiments are conducted on the yeast ortholog of RBM28, Nop4. They show that mutant Nop4 causes a growth defect in yeast traced down to impaired rRNA processing. They further show that the mutant protein fails to interact with a number of its binding partners. The mutation is in RRM3 of the protein. This domain previously thought to bind RNA is a protein-protein interaction site. Finally, they show that the Leu->Pro mutation disrupts the structural integrity of RRM3 The work is clearly presented and provides new insight into ANE syndrome. The quality and impact of the work make it suitable for *eLife*.

*Reviewer #2:*

The manuscript by McCann et al. entitled "The Molecular Basis for ANE Syndrome Revealed by the Large Ribosomal Subunit Processome Interactions" describes an experimental approach that uses modeling in budding yeast to probe the molecular defects that underlie ANE syndrome. In humans, this disease is caused by a L->P amino acid substitution in the RBM28 protein. The budding yeast orthologue of RBM28, Nop4, has a conserved leucine that is located in the analogous domain as the change detected in patients.

As RBM28 can replace the function of the essential Nop4 protein, the authors modeled the disease-causing amino acid substitution in the budding yeast Nop4 protein. The authors use a nice complement of functional studies coupled with dissection of the key interacting domain and finally structural studies. The study is fairly straightforward but does provide insight into what mechanisms might underline disease and also provides insight into key interactions required for proper large subunit assembly.

The authors make the surprising finding that RRM3 and 4 of Nop4 are sufficient to confer function and they seem to do so by mediating protein-protein interactions. One experimental issue is that Nop4 functions within the nucleolus and thus needs to be targeted there to function. A quick perusal of the literature shows no published information on how Nop4 enters the nucleus – which it must do to access the nucleolus. There are a number of predicted classical nuclear localization motifs and most are in the C-terminal domain of the protein. While this is not an issue for the two-hybrid system which presumably contains a nuclear targeting module, this could be an issue for the functional studies. The authors should really include localization as an aspect of their studies to validate their conclusions that the Rrm3-4 mediates the essential function of Nop4 and that Rrm1-2 is not sufficient to do so (see detailed comments below).

With the broad interactions examined, the use of the two-hybrid is quite reasonable; however, for at least some of the interactions, the authors should validate the loss of interaction with a biochemical approach. They could easily choose binding partners for which antibodies are available or employ commercially available TAP-tagged proteins.

One minor issue that should be clarified is the reference to the Leucine to Proline amino acid substation as a point mutation. While this change in the codons could easily occur due to a single point mutation in the DNA that changes a leucine codon to a proline codon (presumably this is what happens in the patients?) – it is the amino acid change that is the consequence of the point mutation that causes the functional change. The authors should merely keep this in mind in finalizing the text.

Overall there are some relatively minor points that could enhance the impact of the studies but the experiments are nicely set up and the data presented support the conclusions drawn.

Specific Comments:

Figure 1 presents the phenotype observed. Figure 1 presents a schematic that compares RBM28 with Nop4. Although the detailed amino acid sequences of these proteins are included as a supplemental figure, providing the sequence of amino acids immediately surrounding the conserved leucine would make the information presented in Figure 1 far more convincing that the leucine is located within a conserved domain – without forcing the reader to refer to the supplemental figure.

Figure 3 employs two-hybrid screening to identify interactions that are lot or maintained with the L306P amino acid change. The approach is reasonable for the broad screening. However, the authors should validate at least a few of these differential interactions through a biochemical approach (co-immunoprecipitation being the logical approach). Such biochemistry would complement the rationale use of the two-hybrid analyses.

Figure 5 assesses the domains of Nop4 required for function. The immunoblot shown in Figure 5 is very difficult to interpret. The RRM1-2 domain appears very degraded. The authors need to provide a different blot showing that at least some protein of the predicted size is generated. Arrows marking the correct bands (corresponding to the predicted size) would help. Addition of molecular weight markers would also help in this and other immunoblots shown. The immunoblot shown in Figure 5 does not really compare to the one shown in Figure 4 (albeit they are different constructs- one FLAG-tagged and one for two-hybrid).

The authors really do not comment on the temperature sensitive rescue of rRNA processing and growth by the Rrm3-4 fragment (Figure 5 and Figure 6). The legend for Figure 6 does not mention the different temperatures. The authors need to provide some insight into why this rescue is temperature-dependent. Do they suspect that the domain may be less stable at higher temperatures? Some immunoblotting carried out from cells grown at the different temperatures might help to address this temperature-sensitive effect.

The other experimental issue that should be addressed for this analysis is the localization of the protein domains. These fragments appear to be FLAG-tagged so it should be straightforward to examine the localization of the protein domains and append an NLS if needed. Could the two-hybrid clones be used to test for function?

It would be nice to have a bit more information about how RBM28 rescues the yeast phenotype and whether the amino acid substitution in the human protein recapitulates the defects seen with the altered yeast protein.

The authors are now in a position to determine which interactions are critical for the function of Nop4. This point could be mentioned in the Discussion.

*Reviewer #3:*

The authors have taken the observation, by others, that a recessive mutation in RBM28, a human homologue of Nop4 of yeast, causes ribosomopathy, and expanded on it with reference to their recent very nice work on the Processosome (McCann et al. G&D). They show that when this mutation is transferred to the yeast gene, the formation of 25S rRNA is partially disrupted, (Figure 2) and that the interaction of Nop4 with a number of other proteins is reduced (Figure 3). These are interesting findings, but perhaps not surprising since the mutation causes substantial reduction of 60S ribosome synthesis in human cells. The rest of the paper is concerned with Nop4 and its fragments in yeast, where they conclude that only the C=terminal half of Nop4 is sufficient for growth, but only at 23 degrees. Quite similar experiments were reported by Sun & Woolford in 1997 (L611) that suggested all 4 RRMs were essential. But the difference is simply a matter of temperatures used for growth. Finally, (Figure 7) some protein structural measurements suggest that the L>P mutation disrupts the structure of RRM3 suggesting that leads to loss of interactions with other members of the processosome.

I have a number of problems with the data, but overall I feel the authors have sold themselves short by not investigating just how the mutation affects 60S ribosome synthesis, which the yeast system is perfect for. Instead they have confined themselves to nibbling around the edges of a problem that they are in the perfect position to solve. While I am not in a position to critique Figure 7, again it seems to me that more could be done, e.g. the effects of mutation & temperature on structure and on interaction with other proteins (Figure 3).

In summary, the data presented are for the most part sound, but the conclusions contribute only marginally to deep analysis of the problem.

*Reviewer #4:*

In this manuscript McCann et al. investigate the molecular basis of ANE syndrome pathogenesis. ANE syndrome is caused by a mutation in the nucleolar protein RBM28 (L351P) which affects the LSU processome resulting in ribosome biogenesis defects. McCann et. al use yeast as a model to study the effect of ANE mutation on the structure and function (as a hub protein) using the homologous protein, Nop4p. The authors first validate the model system, demonstrate that the ANE mutation affects Nop4 "hub" function, investigate the molecular biology of the RRM3 and RRM4 domains of Nop4 and conclude with biophysical analyses demonstrating that the L351P mutation disrupts folding of the RRM3 domain. This is a very straightforward study employing basic yeast molecular genetics and complemented with some nice biophysical studies. The conclusions are warranted by the data. However, the work could benefit from some minor improvement, and some of the data analyses are questionable.

Figure 1.

The abbreviation EV is not explained/mentioned in the figure legend and text. Figure 1 is missing quantification. This is important because Nousbeck et. al 2008 reported that decreased expression of mutant RBM28 in ANE patient cells. Also, the presence of additional bands in the anti-FLAG and anti-Mpp10 blots are not explained.

The growth defect for the Nop4 L306P expressing strain is described as 'severe' as compared to the EV expressing strain (for 1D and 1E). However, the EV expressing strain is essentially "dead' (in 1D). Thus, the proper comparison should be of the mutant to the WT. This renders the growth defect 'moderate'.

The growth assays in Figure 1 have some issues: a) the time points are randomly selected (more time points are clustered around 20 hrs). b) the data could be quantified, i.e. doubling times should be calculated. c) While the EV expressing strain is dead in the dilution spot assays, it seems to have grown to in the liquid media. This can be explained as a consequence of residual Nop4 after gene shutoff.

Figure 1—figure supplement 3: While it is clear that the authors did identify the correct yeast leucine residue to mutate, the use of a simple 1 to 1 protein alignment also gives the impression of someone having done the minimal effort. Additional alignments and phylogenetic analyses are simple to perform, and would lend more credibility to the choice of which leucine to mutate.

Figure 2.

There are some significant problems with this figure, especially as it compares with Figure 6. In general, it appears that Figure 2 and Figure 6 were performed by different people at different times each having different standards for quality and data interpretation.

In the text, the reference to Figure 2 in the first paragraph of the subsection “The ANE syndrome mutation causes pre-rRNA processing defects in yeast” is confusing. The text would seem to indicate that the figure shows that "…the mature 25S rRNA and the 27S and 7S pre-rRNA precursors are severely reduced in yeast depleted of Nop4". However, Figure 2 only shows the processing schema.

Figure 2 top panel (gel) uses Ethidium bromide to quantify the 25S and 18S rRNA. The method is not terribly sensitive for quantitation. To be honest, I cannot discern the differences between the wild-type and mutant in either the top or bottom panels of Figure 2 that are graphed in panel C. Additionally, in the 48 hr EV lane, there is so little rRNA (because the cells are dead) that it borders on disingenuous to claim any quantitation in Panel C. Compare these gels/autorads to those shown in Figure 6: these ones are much less informative.

Unlike Figure 6, Figure 2 is missing a loading control.

The claim that Nop4 L306P results in 'severe reduction in 27S and 7S levels…' (subsection “The ANE syndrome mutation causes pre-rRNA processing defects in yeast”, last paragraph) is not supported by the quantification in 2C. The fold change as compared to WT is moderate.

The claim that the EV control had most severe reduction in 25S/18S ratio and also 27S and 7S levels (subsection “The ANE syndrome mutation causes pre-rRNA processing defects in yeast”) again borders on disingenuous, because this mutant is dead, or at the very least, in the process of dying.

The statistical comparisons in Figure 2 are different than those in Figure 6. Here, comparisons are performed with respect to wild-type (which is correct). In 6B, the comparisons are with respect to empty vector (which is wrong, again because EV is dead).

Figure 3.

It is unclear why such a small panel of bait reporters were assayed here, especially in light of the fact that the lab possesses 23 bait reporters already (as shown in Figure 4). Again, were Figure 3 and Figure 4 done by different people? Assaying the entire panel would illuminate the role of Nop4 in the interactome and the consequences of the mutation.

Including the Nop4 interactome map of as an additional panel (from McCann, K. L et.al 2015. Genes Dev) might be helpful in illuminating which partners, and thus which pathways in ribosome biogenesis, may be affected by this mutation.

Figure 4.

Figure 4. Western lot analysis should be accompanied with a quantification graph. Also, compare this to Figure 5. Why are they so different?

Figure 4. Since the mutation associated with ANE is in RRM3, one would like to see a construct expressing RRM1, 2 and 4 only. This would be a great negative control.

The protocol notes that the growth time for the Y2H assay in 4C was 2 weeks. Those familiar with Y2H assays might be concerned about the high false positive rate when these assays are performed in the cold for such a long time.

Co-immunoprecipitation assays would serve as orthogonal test of the protein -protein interactions.

Figure 5.

Figure 5, lanes 2 and 3 have a lot of background. Additionally, the figure lacks quantification.

Again the growth assays in 5C and 5D a) contain random time points (time points not equally spaced) b) missing quantification of doubling time and c) the EV control is dead in the dilution spot assays, however it grows in liquid media.

In Figure 5 the scale on Y-axis is misleading. It seems that the log scale on Y-axis starts at 'zero' because the origin is not labeled. The next tick upward on the Y-axis is also not labeled.

Figure 6.

Again, Figure 6 top panel (gel) uses Ethidium bromide to quantify the 25S and 18S rRNA. However, here, we can actually see a difference at 30°C.

In 6B the significance of the ratios 25S/18S, 35S/Scr1, 27S/Scr1 and 7S/Scr1 is calculated as compared to the EV control. However, EV control strain is dead. As noted above, the proper comparison should be to WT.

There is no way to compare the quantitative graphs in Figure 2 with Figure 6 because Figure 2 lacks a loading control. This results in an apples to oranges comparison that only serves to confuse the reader. For example, the claim that RRM 3-4 significantly restored the pre-rRNA processing defects at 23°C is supported by the data in 6B as there is a discernable difference in the ratios. However, the similar analyses shown in Figure 2 indicate that the differences are lesser, but the claim is that the effects are greater. I am totally confused.

Figure 7.

This is the biophysical characterization of the mutant, but it is done with the RRM3 fragment of the human protein. While I do not have a problem with the switch from yeast-based studies to the human protein, an explanation for why would be informative. (Indeed, one might ask, if the human protein can complement deletion of Nop4 in yeast, why weren't these studies performed using the human protein to begin with?). Figure 7 (NMR data) requires more explanation to illuminate the differences between the WT and the mutant protein structure. As written, the information content is minimal.

Figure 7 needs labels: a) WT and mutant protein and b) location of the mutation L351P.

*Reviewer #4 (Additional data files and statistical comments):*

As noted in the long form review, quantitative analyses are lacking in some places, and the statistical analyses performed in Figure 2 and Figure 6 need to be aligned.

---

## [Author Response]

[Editors’ note: the author responses to the first round of peer review follow.]

Reviewer #2:

*The authors make the surprising finding that RRM3 and 4 of Nop4 are sufficient to confer function and they seem to do so by mediating protein-protein interactions. One experimental issue is that Nop4 functions within the nucleolus and thus needs to be targeted there to function. A quick perusal of the literature shows no published information on how Nop4 enters the nucleus – which it must do to access the nucleolus. There are a number of predicted classical nuclear localization motifs and most are in the C-terminal domain of the protein. While this is not an issue for the two-hybrid system which presumably contains a nuclear targeting module, this could be an issue for the functional studies. The authors should really include localization as an aspect of their studies to validate their conclusions that the Rrm3-4 mediates the essential function of Nop4 and that Rrm1-2 is not sufficient to do so (see detailed comments below).*

Reviewer #2 makes an excellent point that the failure to rescue by RRM1-2 may simply be due to a mislocalization of this fragment. To address this concern, we repeated the growth complementation assays in Figure 5 using the yeast two-hybrid vector, pACT2, which will ensure nuclear localization. We found that even when targeted to the nucleus, RRM1-2 fails to rescue the growth defect conferred by depletion of endogenous Nop4 in both solid and liquid medium. Additionally, using immunofluorescence, we confirmed the nuclear localization of the Nop4 fragments expressed from the yeast two-hybrid vector, pACT2. These results have been included in the Results section and have been added to Figure 5—figure supplement 1.

With the broad interactions examined, the use of the two-hybrid is quite reasonable; however, for at least some of the interactions, the authors should validate the loss of interaction with a biochemical approach. They could easily choose binding partners for which antibodies are available or employ commercially available TAP-tagged proteins.

We have now done this and included the results in new panels of Figure 3. To validate the loss of interaction in the presence of the ANE syndrome mutation, we employed the co-immunoprecipitation approach described in (McCann et al. 2015, Genes & Dev). This assay utilizes yeast expression vectors that express 3xHA or 3xFLAG tagged proteins of interest. We prefer not to use commercially available strains or antibodies as we find that making them ourselves provides greater experimental reliability. We assayed for the loss of interaction between Nop4 and Noc2, Mak5, Nop4, Nsa2 and Dbp10 when the ANE syndrome mutation (L306P) is present. We found that the co-immunoprecipitation results mirrored the yeast two-hybrid results. In the presence of the ANE syndrome mutation (L306P), the amount of co-purifying Nop4 was significantly reduced when immunoprecipitating with Noc2, Mak5, Nop4 and Nsa2 but was not significantly affected when immunoprecipitating with Dbp10. These results are discussed in the Results section and have been added to new panels in Figure 3.

One minor issue that should be clarified is the reference to the Leucine to Proline amino acid substation as a point mutation. While this change in the codons could easily occur due to a single point mutation in the DNA that changes a leucine codon to a proline codon (presumably this is what happens in the patients?) – it is the amino acid change that is the consequence of the point mutation that causes the functional change. The authors should merely keep this in mind in finalizing the text.

Thank you for this important comment. We have edited the text such that we refer to the Leucine to Proline mutation as an amino acid substitution rather than as a point mutation.

*Overall there are some relatively minor points that could enhance the impact of the studies but the experiments are nicely set up and the data presented support the conclusions drawn. Specific Comments: Figure 1 presents the phenotype observed. Figure 1 presents a schematic that compares RBM28 with Nop4. Although the detailed amino acid sequences of these proteins are included as a supplemental figure, providing the sequence of amino acids immediately surrounding the conserved leucine would make the information presented in Figure 1 far more convincing that the leucine is located within a conserved domain – without forcing the reader to refer to the supplemental figure.* We have now done this. Figure 1 has been changed to include the amino acid alignment of the ~30 amino acids of RRM3 surrounding the conserved leucine that is mutated in ANE syndrome. The alignment has also been expanded to include four other species.

Figure 3 employs two-hybrid screening to identify interactions that are lot or maintained with the L306P amino acid change. The approach is reasonable for the broad screening. However, the authors should validate at least a few of these differential interactions through a biochemical approach (co-immunoprecipitation being the logical approach). Such biochemistry would complement the rationale use of the two-hybrid analyses.

We have now done this as stated above. We validated the loss of interaction in the presence of the ANE syndrome mutation, through the co-immunoprecipitation approach described in (McCann et al. 2015, Genes & Dev). We assayed for the loss of interaction between Nop4 and Noc2, Mak5, Nop4, Nsa2 and Dbp10 when the ANE syndrome mutation (L306P) is present. We found that significantly less Nop4 L306P was co-purified with Noc2, Mak5, Nop4 and Nsa2. These data are included in the Results and has been added to Figure 3.

*Figure 5 assesses the domains of Nop4 required for function. The immunoblot shown in Figure 5 is very difficult to interpret. The RRM1-2 domain appears very degraded. The authors need to provide a different blot showing that at least some protein of the predicted size is generated. Arrows marking the correct bands (corresponding to the predicted size) would help. Addition of molecular weight markers would also help in this and other immunoblots shown. The immunoblot shown in Figure 5 does not really compare to the one shown in Figure 4 (albeit they are different constructs- one FLAG-tagged and one for two-hybrid).* We have replaced the blot in Figure 5 and added molecular weight markers. The predicted sizes of the different domains have been added to the figure legend. Additionally, we have added either molecular weight markers or arrows marking the correct bands to all the western blots in the manuscript.

The authors really do not comment on the temperature sensitive rescue of rRNA processing and growth by the Rrm3-4 fragment (Figure 5 and Figure 6). The legend for Figure 6 does not mention the different temperatures. The authors need to provide some insight into why this rescue is temperature-dependent. Do they suspect that the domain may be less stable at higher temperatures? Some immunoblotting carried out from cells grown at the different temperatures might help to address this temperature-sensitive effect.

We have added the temperature and length of depletion to the figure legend. At present we cannot speculate why rescue would work better at 30°C and 23°C than at 37°C though a likely explanation, as the reviewer points out, is that at higher temperatures the domain folds less well.

*The other experimental issue that should be addressed for this analysis is the localization of the protein domains. These fragments appear to be FLAG-tagged so it should be straightforward to examine the localization of the protein domains and append an NLS if needed. Could the two-hybrid clones be used to test for function?* As we stated above, we have addressed this concern by repeating the growth complementation assays in Figure 5 using the yeast two-hybrid vector, pACT2, which will ensure nuclear localization. We found that even when targeted to the nucleus, RRM1-2 fails to rescue the growth defect conferred by depletion of endogenous Nop4. Furthermore, we confirmed the nuclear localization of the Nop4 fragments expressed from the yeast two-hybrid vector, pACT2, using immunofluorescence. These data have been included in the Results section and have been added to Figure 5—figure supplement 1.

*It would be nice to have a bit more information about how RBM28 rescues the yeast phenotype and whether the amino acid substitution in the human protein recapitulates the defects seen with the altered yeast protein.* Although it has been shown by others that RBM28 can complement the growth defect upon inactivation of Nop4 (Kachroo et al. Science 2015), we have repeated the complementation experiment using a strain where we can conditionally deplete Nop4 by changing the carbon source in the growth medium from galactose to glucose (dextrose) (Figure 1). We have discussed these new results in the Results section and included them in Figure 1—figure supplement 2. As previously reported in Kachroo et al., we found that RBM28 was able to complement the growth defect observed upon depletion of Nop4 at 37 °C, 30 °C and 23 °C, but not at 17°C, on solid medium.

We chose not to study the mutated human protein (RBM28 L351P) in yeast because the mutation would be studied out of its natural context. Determining the effects of the human mutation in a more relevant experimental system, such as human cell culture or *Xenopus tropicalis*, both of which we have used previously to study ribosomopathies (Griffin et al. PLoS Geneti. 2015, Freed et al. PLoS Geneti. 2015), is an important future goal for my laboratory.

*The authors are now in a position to determine which interactions are critical for the function of Nop4. This point could be mentioned in the Discussion.* We have added this to the Discussion.

Reviewer #3:

*The authors have taken the observation, by others, that a recessive mutation in RBM28, a human homologue of Nop4 of yeast, causes ribosomopathy, and expanded on it with reference to their recent very nice work on the Processosome (McCann et al. G&D). They show that when this mutation is transferred to the yeast gene, the formation of 25S rRNA is partially disrupted, (Figure 2) and that the interaction of Nop4 with a number of other proteins is reduced (Figure 3). These are interesting findings, but perhaps not surprising since the mutation causes substantial reduction of 60S ribosome synthesis in human cells. The rest of the paper is concerned with Nop4 and its fragments in yeast, where they conclude that only the C=terminal half of Nop4 is sufficient for growth, but only at 23 degrees. Quite similar experiments were reported by Sun & Woolford in 1997 (L611) that suggested all 4 RRMs were essential. But the difference is simply a matter of temperatures used for growth. Finally, (Figure 7) some protein structural measurements suggest that the L>P mutation disrupts the structure of RRM3 suggesting that leads to loss of interactions with other members of the processosome.* With all due respect, the reviewer is mistaken that we know that “the mutation causes substantial reduction of 60S ribosome synthesis in human cells.” Indeed, this is not known, either for human or yeast cells. Our work here is the first time the consequences of the mutation have been investigated experimentally in either organism. In Nousbeck et al. (2008) where the ANE syndrome was first described, the authors counted mature free cytoplasmic ribosomes in micrographs of cultured fibroblasts by transmission EM, and found that numbers were reduced in patients with ANE syndrome. That is the extent of the experiment: no analysis of 40S or 60S synthesis was done. So we do not yet know how the ANE syndrome mutation affects ribosome biogenesis in any organism. That is one reason why our work presented here is so important and highly significant.

In addition, it is not true that “quite similar experiments were reported by Sun & Woolford in 1997.” While it is true that they made a missense mutation in RRM3, this mutation was at a different amino acid than the ANE syndrome mutation. They did not examine the effects of the ANE syndrome mutation, as it had not been discovered yet.

*I have a number of problems with the data, but overall I feel the authors have sold themselves short by not investigating just how the mutation affects 60S ribosome synthesis, which the yeast system is perfect for. Instead they have confined themselves to nibbling around the edges of a problem that they are in the perfect position to solve. While I am not in a position to critique Figure 7, again it seems to me that more could be done, e.g. the effects of mutation & temperature on structure and on interaction with other proteins (Figure 3). In summary, the data presented are for the most part sound, but the conclusions contribute only marginally to deep analysis of the problem.* We are concerned that the reviewer has read our manuscript with the mistaken impression that much of the work has previously been done, and therefore that our work does not provide much new insight. We vigorously disagree.

First, as discussed above, the ANE syndrome mutation has not been studied at the biochemical level before in any organism. The results from patient material presented in Nousbeck et al. indicate reduced levels of free cytoplasmic ribosomes by counting them in electron micrographs. This is a non-biochemical and somewhat crude level of analysis that was probably a result of the limitations of obtaining patient material. This study here is the first to define the consequences of the ANE syndrome mutation in any organism, and therefore is valuable for furthering our understanding of this ribosomopathy, and of ribosome biogenesis in general.

Second, introducing the orthologous amino acid change that causes ANE syndrome into yeast Nop4, we were able to pinpoint its effects on growth and pre-rRNA processing, and to define precisely how defective the mutated Nop4 is for the first time. Similarly, these experiments have not been done or published before in any organism. It is critical for understanding the molecular basis of ANE syndrome to determine the extent to which the mutated protein causes defective pre- rRNA processing, as we have done.

Third, based on our recent published work finding that Nop4 is a hub protein in the LSU processosome interactome (McCann et al. Genes Dev 2015), we have gone one to show unexpected and surprising aspects of the function of the RRMs in Nop4. While RRMs are named for being RNA binding domains, we find here that instead they play a critical role in protein interaction. Introduction of the ANE syndrome mutation into RRM3 reduces interaction with a subset of proteins, as we have now shown by two methods. Furthermore, RRMs 3-4 are sufficient for protein interaction and growth, a truly novel finding.

We do not understand how this extensive work that required 7 figures with an average of about 4 parts, all of which are new experiments that have never been done before and that provide a molecular basis for the ANE syndrome for the first time, can be considered “nibbling around the edges of a problem.” The reviewer writes that “the authors have sold themselves short by not investigating just how the mutation affects 60S ribosome synthesis.” But this is exactly what we have done.

Reviewer #4:

*In this manuscript McCann* et al.

*investigate the molecular basis of ANE syndrome pathogenesis. ANE syndrome is caused by a mutation in the nucleolar protein RBM28 (L351P) which affects the LSU processome resulting in ribosome biogenesis defects. McCann et. al use yeast as a model to study the effect of ANE mutation on the structure and function (as a hub protein) using the homologous protein, Nop4p. The authors first validate the model system, demonstrate that the ANE mutation affects Nop4 "hub" function, investigate the molecular biology of the RRM3 and RRM4 domains of Nop4 and conclude with biophysical analyses demonstrating that the L351P mutation disrupts folding of the RRM3 domain. This is a very straightforward study employing basic yeast molecular genetics and complemented with some nice biophysical studies. The conclusions are warranted by the data. However, the work could benefit from some minor improvement, and some of the data analyses are questionable. Figure 1.*

The abbreviation EV is not explained/mentioned in the figure legend and text.

We are sorry that this was not clear enough. While, the abbreviation EV is explained in the Results and in the legend for Figure 3 we did not explain it consistently in the entire manuscript. We have now edited the text such that EV is defined in every section of the Results and in every figure legend, as requested.

Figure 1 is missing quantification. This is important because Nousbeck et. al 2008 reported that decreased expression of mutant RBM28 in ANE patient cells. Also, the presence of additional bands in the anti-FLAG and anti-Mpp10 blots are not explained.

While the reviewer makes an excellent point that Nousbeck et al. observed a substantial difference in expression of RBM28 in the human ANE patient cells, this experiment is different. Both Nop4 WT and Nop4 L306P are being constitutively expressed from a highly expressing promoter on a plasmid. Thus, we would expect them to be expressed at fairly similar levels, which is what we observe when we quantitate the bands. We have now included the quantitation in the figure legend. Additionally, we have labeled the extra bands in the figure and explained them in the figure legend.

The growth defect for the Nop4 L306P expressing strain is described as 'severe' as compared to the EV expressing strain (for 1D and 1E). However, the EV expressing strain is essentially "dead' (in 1D). Thus, the proper comparison should be of the mutant to the WT. This renders the growth defect 'moderate'.

We are sorry to disagree with this reviewer but this is factually incorrect. While the reviewer claims that the growth defect for Nop4 L306P is described as “severe as compared to the EV expressing strain,” the text actually reads: “The L306P mutation impaired growth at all temperatures tested compared to WT, although the defect was not as severe as that observed with the EV control (Figure 1).” We agree with this reviewer that the growth defect was moderate which is also consistent with our conclusion that the ANE syndrome mutation is a hypomorphic allele.

The growth assays in Figure 1 have some issues: a) the time points are randomly selected (more time points are clustered around 20 hrs).

The growth curve in Figure 1 has been replaced with a growth curve with more time points that are less clustered around 20 hours. It is important to note that the result is the same even though there are more time points and they are less clustered.

b) The data could be quantified, i.e. doubling times should be calculated.

Accurate doubling times cannot be calculated from this type of experiment because the growth rate changes over time. However, we have estimated the doubling times based on the new growth curve in Figure 1. The doubling times have been included in both the figure and the Results.

*c) While the EV expressing strain is dead in the dilution spot assays, it seems to have grown to in the liquid media. This can be explained as a consequence of residual Nop4 after gene shutoff.* The reviewer is correct that the empty vector (EV) expressing strain does not grow in the dilution spot assays but does grow in liquid media. The EV strain is able to grow in liquid media because the GAL promoter is known to be leaky (Park et al. Yeast 2011). Furthermore, growth in liquid medium is quantitative and allows for the estimation of doubling times. This is why we assay growth both ways.

Figure 1—figure supplement 3: While it is clear that the authors did identify the correct yeast leucine residue to mutate, the use of a simple 1 to 1 protein alignment also gives the impression of someone having done the minimal effort. Additional alignments and phylogenetic analyses are simple to perform, and would lend more credibility to the choice of which leucine to mutate.

We have repeated the alignment analyses with more species included rhesus (M. mulatta), mouse (M. musculus), frogs (X. tropicalis) and fish (D. rerio). We have edited Figure 1 such that it now includes the expanded amino acid alignment of the ~30 amino acids of RRM3 surrounding the conserved leucine that is mutated in ANE syndrome.

Additionally, we have added a new supplemental figure (new Figure 1—figure supplement 1) that is the full alignment of RRM3 from humans, yeast, rhesus, mouse, frogs and fish.

From these expanded analyses, we can confidently conclude that we had identified and mutated the correct leucine in yeast Nop4 as the reviewer points out.

Figure 2.

There are some significant problems with this figure, especially as it compares with Figure 6. In general, it appears that Figure 2 and Figure 6 were performed by different people at different times each having different standards for quality and data interpretation.

*In the text, the reference to Figure 2 in the first paragraph of the subsection “The ANE syndrome mutation causes pre-rRNA processing defects in yeast” is confusing. The text would seem to indicate that the figure shows that "…the mature 25S rRNA and the 27S and 7S pre-rRNA precursors are severely reduced in yeast depleted of Nop4". However, Figure 2 only shows the processing schema.* Yes, the reviewer is absolutely correct that we reference the processing schema in Figure 2. We were trying to help the reader understand the complexity of pre-rRNA processing and the nature of the Nop4 depletion phenotype by pointing them to a diagram. As the Nop4 depletion processing phenotype has already been published, we included the references to the relevant papers (Sun and Woolford EMBO 1994, Berges et al. EMBO 1994). However, to be more clear, we have also included a reference to the northern blots in Figure 2, which show the same defect.

Figure 2 top panel (gel) uses Ethidium bromide to quantify the 25S and 18S rRNA. The method is not terribly sensitive for quantitation. To be honest, I cannot discern the differences between the wild-type and mutant in either the top or bottom panels of Figure 2 that are graphed in panel C. Additionally, in the 48 hr EV lane, there is so little rRNA (because the cells are dead) that it borders on disingenuous to claim any quantitation in Panel C. Compare these gels/autorads to those shown in Figure 6: these ones are much less informative.

Ethidium bromide is commonly used to visualize and quantify the mature rRNAs in both yeast and human cell systems (Tafforeau et al. Mol Cell 2013, Freed and Baserga NAR 2010, Fatica et al. MCB 2003, Tollervey et al. Cell 1993, Russell and Tollervey J. Cell Biol. 1992).

We have included Author response Table 1 and Figure 8 to substantiate our claim that the ANE syndrome mutation disrupts processing of the 25S based on the ethidium bromide staining and subsequent quantitative analysis of the mature 25S and 18S rRNAs. Author response Table 1 contains the quantitative analysis of the 25S and 18S rRNAs by ethidium bromide staining. Image J is much more sensitive than the naked eye and was able to detect signal from the empty vector (EV) lane. Figure 8 includes a longer exposure of the ethidium bromide stained gel from Figure 2. In this longer exposure, the 25S rRNA is much more visible (Figure 8). However, we chose not to use this particular image in the figure because the other lanes are significantly overexposed. Additionally, we have included an image of one methylene blue stained membrane that was subsequently used for northern blot analysis (Figure 8 left panel). Methylene blue also detects the mature 25S and 18S rRNAs. We have repeated the quantitative analysis of the 25S and 18S rRNAs and obtained the same result (Figure 8 right panel).

We are sorry that the reviewer does not discern a difference between the wild-type and the mutant by eye. This is why it is so important to include a quantitative analysis. We agree with the reviewer that the differences are subtle, which is consistent with the ANE syndrome mutation being a hypomorphic allele. However, as we have demonstrated, there is a method independent, reproducible, statistically significant difference between 25S/18S ratios for the wild-type and the ANE syndrome Nop4 mutant protein.

Author response Table 1: ethidium bromide raw data and quantitation

25S18S25S/18S25S/18S NormEV 06259.44710810.880.5789951.115136709EV 08290.41810043.830.8254241.058714794EV 08786.4476811.0331.2900311.238299455EV 05530.1548089.9830.683580.916887383EV 48201.021986.2340.2038270.18200229EV 48795.7492513.8610.3165450.283377889EV 48616.7991853.6690.3327450.150671679EV 48316.6071720.5980.184010.188479377Nop4 WT 04093.647884.2960.5192141Nop4 WT 07770.0049966.0540.7796471Nop4 WT 04694.1544505.9121.0417771Nop4 WT 04193.2765624.4470.7455451Nop4 WT 4810630.429492.1751.1199141Nop4 WT 488969.2968029.5181.117041Nop4 WT 4810882.324927.6692.2084111Nop4 WT 488543.3478750.8610.9762861Nop4 L306P 04376.9838405.1750.5207491.002954827Nop4 L306P 07532.9839358.9330.8048981.032387275Nop4 L306P 05838.3474754.1541.2280521.178805166Nop4 L306P 06387.3977497.690.8519151.142675222Nop4 L306P 481967.6696570.5390.2994680.267403102Nop4 L306P 481861.0123429.5980.5426330.485777142Nop4 L306P 489506.0546683.791.4222550.644017451Nop4 L306P 487573.76111640.350.6506470.666451283

Author response image 1.The ANE syndrome mutation (L306P) disrupts 25S production in yeast.(**A**) Left panel: longer exposure of the ethidium bromide stained gel in Figure 2. Total RNA was extracted from yeast expressing no Nop4 (EV), Nop4 WT or Nop4 L306P after depletion of endogenous Nop4 for the indicated time. Right panel: The ratios of the mature rRNAs (25S/18S) were calculated from four replicate EtBr experiments and were plotted with error bars representing the standard deviation. Significance compared to WT was evaluated using one-way ANOVA. (**B**) Left panel: Methylene blue stained membrane of the northern blot in Figure 2. Total RNA was extracted from yeast expressing no Nop4 (EV), Nop4 WT or Nop4 L306P after depletion of endogenous Nop4 for the indicated time. Right panel: The ratios of the mature rRNAs (25S/18S) were calculated from four replicate methylene blue experiments and were plotted with error bars representing the standard deviation. Significance compared to WT was evaluated using one-way ANOVA. **** indicates a p value <0.0001. ***indicates a p value <0.001.**DOI:**
http://dx.doi.org/10.7554/eLife.16381.019

Unlike Figure 6, Figure 2 is missing a loading control.

To address this concern, we have re-probed the relevant northern blots with the loading control, Scr1. Additionally, we have updated our analysis of the data to include the ratios of the individual pre-rRNA species to the loading control. The Scr1 loading control blots and the additional quantitative analyses are now included in Figure 2. We have also edited the Results to include this new data. The quantitation of the levels of the individual pre-rRNA species compared to the loading control further supports our claim that the ANE syndrome mutation disrupts processing of the large subunit ribosomal RNAs, as there is a subtle but statistically significant decrease of the 7S pre-rRNA in the presence of the ANE syndrome mutation.

The claim that Nop4 L306P results in 'severe reduction in 27S and 7S levels…' (subsection “The ANE syndrome mutation causes pre-rRNA processing defects in yeast”, last paragraph) is not supported by the quantification in 2C. The fold change as compared to WT is moderate.

We are sorry to disagree with this reviewer but this is factually incorrect. In the text, we wrote: “Depletion of Nop4 resulted in a severe reduction of 27S and 7S levels, with a concomitant decrease in the 27S/35S and 7S/35S ratios, indicative of an ITS1 processing defect, as has been previously observed (Figure 2; Bergès et al. 1994, Sun and Woolford 1994). The Nop4 L306P mutant showed an intermediate growth defect and also displayed an intermediate, but statistically significant ITS1 processing defect as indicated by reduced 27S/35S and 7S/35S ratios (Figure 2).” We did not claim that the mutation resulted in a severe reduction. To attempt to make this more clear, we have rewritten this section.

The claim that the EV control had most severe reduction in 25S/18S ratio and also 27S and 7S levels (subsection “The ANE syndrome mutation causes pre-rRNA processing defects in yeast”) again borders on disingenuous, because this mutant is dead, or at the very least, in the process of dying.

An empty vector (EV) is frequently used in my laboratory and in others as a null control in experiments to ascertain the affect of depletion or mutation on yeast growth and/or ribosome biogenesis (Ferreira-Cerca et al. NAR 2014, Qiu et al. PNAS 2014, Tomecki et al. EMBO 2010, Freed and Baserga NAR 2010, Bohnsack et al. Mol Cell 2009, Bohnsack et al. EMBO 2008, Leulliot et al. NAR 2007, Bleichert et al. PNAS 2006, Kos and Tollervey Mol Cell 2005). Nop4 is essential and yeast depleted of Nop4 (EV null control) will eventually die. However, as the growth curves in Figure 1 and Figure 5 demonstrate, we harvested RNA while the EV control strain was still doubling.

The statistical comparisons in Figure 2 are different than those in Figure 6. Here, comparisons are performed with respect to wild-type (which is correct). In 6B, the comparisons are with respect to empty vector (which is wrong, again because EV is dead).

The statistical comparisons in Figure 2 and Figure 6 are different because the experiments, and therefore, the null hypotheses, are different. In Figure 2, the experiment was to test whether the presence of the mutation disrupts Nop4 function. The null hypothesis is that the mutation does not alter Nop4 function. The expectation is that the mutant will behave like the wild type. Therefore, the statistical analyses were done comparing the mutant to wild type.

In Figure 6, the experiment was to test whether a fragment of Nop4 can complement the pre-rRNA processing defect due to Nop4 depletion. The null hypothesis is that the fragments will not complement. The expectation, then, is that the fragments will behave like the empty vector control (no complementation). Thus, the statistical analyses were done comparing the fragments to the empty vector.

Figure 3.

It is unclear why such a small panel of bait reporters were assayed here, especially in light of the fact that the lab possesses 23 bait reporters already (as shown in Figure 4). Again, were Figure 3 and Figure 4 done by different people? Assaying the entire panel would illuminate the role of Nop4 in the interactome and the consequences of the mutation.

Figure 3 was intended to demonstrate that the presence of the ANE syndrome mutation (L306P) disrupts some, but not all, protein-protein interactions. The reviewer does make a good point that assaying the entire panel would be important for thoroughly mapping the interactions that are disrupted by the mutation. We plan to complete the yeast two-hybrid with the entire panel of bait reporters as part of a future manuscript.

Including the Nop4 interactome map of as an additional panel (from McCann, K. L et.al 2015. Genes Dev) might be helpful in illuminating which partners, and thus which pathways in ribosome biogenesis, may be affected by this mutation.

Copyright prohibits us from re-publishing this figure. We made a table and have now included it in the supplementary material for the reviewer’s convenience.

Figure 4.

Figure 4. Western lot analysis should be accompanied with a quantification graph. Also, compare this to Figure 5. Why are they so different?

We have quantitated the bands and have updated the figure legend to include this information. While the western blot in Figure 5 has been replaced with a cleaner image (with quantitation), both the original blot and the new blot in 5A show the same trends as seen in the western blot in Figure 4: Nop4 RRM 1-2 is expressed at much higher levels than Nop4 WT while Nop4 RRM 3-4 is expressed at lower levels than Nop4 WT. It is not clear why there are such dramatic differences in expression. However, since the trends remain the same across different expression vectors (pACT in Figure 4 and p414GPD 3xFLAG in Figure 5), this result is likely due to differences in the function, localization and/or stability of the fragments.

Figure 4. Since the mutation associated with ANE is in RRM3, one would like to see a construct expressing RRM1, 2 and 4 only. This would be a great negative control.

We agree with the reviewer that it is critical to determine how the individual RRMs in Nop4 contribute to Nop4 function. It has been shown previously that all 4 RRMs are essential for Nop4 function in vivo. More specifically, mutations to highly conserved amino acids within the individual RRMs of Nop4 were all temperature sensitive and all perturbed assembly of the large ribosomal subunit (Sun and Woolford JBC 1997). However, the precise function of each RRM of Nop4 has yet to be determined. This is an important future goal for my laboratory.

For this manuscript, we did generate a construct expressing only RRM3 and attempted to determine if RRM3 alone was sufficient to mediate protein-protein interactions by yeast two-hybrid. However, RRM3 alone is not expressed and therefore its function could not be assayed in any experiments. This has been added to the Results.

The protocol notes that the growth time for the Y2H assay in 4C was 2 weeks. Those familiar with Y2H assays might be concerned about the high false positive rate when these assays are performed in the cold for such a long time.

We are confused about why the reviewer thought we did the experiment at 4°C, where yeast will not grow, as this was not written in the text. We are also confused about the literature that the reviewer draws on about the high false positive rates for Y2H experiments performed in the cold. We are not aware of such a literature and we have been doing Y2H experiments since 1999 (Lee & Baserga 1999 MCB).

As with other, published Y2H experiments from us and others (McCann 2015 Genes & Development, Rolland 2014 Cell, Hegele 2012 Mol. Cell, Charette and Baserga 2010 RNA, Freed and Baserga 2010 NAR, Wong 2007 MBC, Ito 2001 PNAS, Drees 2001 JCB, Uetz 2000 Nature), the Y2H experiments in this manuscript were always performed at 30°C. This was included in the methods for the ANE syndrome Y2H experiment but had not been included for the fragment Y2H experiment. The published screen (McCann 2015 Genes & Development) was actually incubated for 3 weeks at 30°C. We have now indicated the 30°C temperature in the methods for this experiment.

Co-immunoprecipitation assays would serve as orthogonal test of the protein -protein interactions.

The use of the yeast two-hybrid assay to identify protein-protein interactions has been validated extensively by our lab and others (McCann 2015 Genes & Development, Hegele 2012 Mol. Cell, Wang 2011 Mol Syst Biol., Suter 2007 Genome Res.). The vast majority of high-confidence interactions identified by yeast two-hybrid are recapitulated by co-immunoprecipitation. All of the Nop4 interactions examined here have been previously validated by co-immunoprecipitation (McCann 2015 Genes & Development). Thus, we feel confident in our use of the yeast two-hybrid assay to identify biologically relevant protein-protein interactions. The yeast two-hybrid assay in Figure 4 was performed twice and the same interactions were observed. Most importantly, there is a striking difference between the number of interactions mediated by Nop4 RRM 1-2 and Nop4 RRM 3-4 despite the fact that Nop4 RRM 1-2 is expressed at higher levels (Figure 4).

Figure 5.

Figure 5, lanes 2 and 3 have a lot of background. Additionally, the figure lacks quantification.

We have replaced the western blot in Figure 5. We have also included a lighter exposure of the Nop4 RRM 1-2 since it is significantly overexpressed in comparison to Nop4 WT and Nop4 RRM 3-4. We have included the quantitation in the figure legend.

Again the growth assays in 5C and 5D a) contain random time points (time points not equally spaced);

We are sorry that the reviewer is unhappy with the spacing of the time points for these growth assays. However, both growth assays have been carried out three times, with each replicate having the same number of time points, and the end result is always the same. Having more or better-spaced time points does not change that.

b) Missing quantification of doubling time;

As we stated above, accurate doubling times cannot be calculated from this type of experiment because the growth rate changes over time. However, we have estimated the doubling times and included them in the figure and the Results.

c) The EV control is dead in the dilution spot assays, however it grows in liquid media.

Again, the reviewer is correct that the empty vector (EV) expressing strain does not grow in the dilution spot assays but does grow in liquid media. This is not a concern for the reasons listed above.

In Figure 5 the scale on Y-axis is misleading. It seems that the log scale on Y-axis starts at 'zero' because the origin is not labeled. The next tick upward on the Y-axis is also not labeled.

We have added the appropriate labels for the Y-axis.

Figure 6.

Again, Figure 6 top panel (gel) uses Ethidium bromide to quantify the 25S and 18S rRNA. However, here, we can actually see a difference at 30°C.

We are glad that the reviewer can visually detect a difference in 25S rRNA levels by ethidium bromide staining. As the quantitation of the ratio of 25S/18S in Figure 2 and Figure 6 demonstrates, the Nop4 RRM 1-2 and Nop4 RRM 3-4 fragments have a much greater reduction in the 25S/18S ratio than the ANE syndrome mutation (L306P), which is why it is easier to observe by eye.

In 6B the significance of the ratios 25S/18S, 35S/Scr1, 27S/Scr1 and 7S/Scr1 is calculated as compared to the EV control. However, EV control strain is dead. As noted above, the proper comparison should be to WT.

We are sorry, but we disagree with the reviewer. This experiment is to test whether a fragment of Nop4 can complement the pre-rRNA processing defect due to Nop4 depletion. In this case, the null hypothesis is that the fragments will not complement, meaning they will behave like the empty vector (EV) control. Thus, the statistical analyses were done comparing the fragments to the empty vector. This is distinctly different than Figure 2 where the null hypothesis is that the mutation does not disrupt Nop4 function and is thereby expected to behave like Nop4 WT. That is why the statistical analyses were done comparing the mutant to the wild type in Figure 2.

There is no way to compare the quantitative graphs in Figure 2 with Figure 6 because Figure 2 lacks a loading control. This results in an apples to oranges comparison that only serves to confuse the reader. For example, the claim that RRM 3-4 significantly restored the pre-rRNA processing defects at 23°C is supported by the data in 6B as there is a discernable difference in the ratios. However, the similar analyses shown in Figure 2 indicate that the differences are lesser, but the claim is that the effects are greater. I am totally confused.

We have included a loading control, Scr1, to Figure 2 and have added the quantitative analysis of the ratios of the pre-rRNA precursors 35S, 27S and 7S to Scr1 in Figure 2. This now allows for a comparison of the data in Figure 2 to the data in Figure 6. However, it is important to remember that while the assays are the same, the questions we are asking in Figure 2 vs. Figure 6 are different.

Additionally, as we stated above, we claimed that the ANE syndrome mutation causes a statistically significant, moderate processing defect. At no point in the manuscript did we compare the data in Figure 2 and Figure 6 and claim that the differences in Figure 2 are greater.

Figure 7.

This is the biophysical characterization of the mutant, but it is done with the RRM3 fragment of the human protein. While I do not have a problem with the switch from yeast-based studies to the human protein, an explanation for why would be informative. (Indeed, one might ask, if the human protein can complement deletion of Nop4 in yeast, why weren't these studies performed using the human protein to begin with?).

We have included a sentence to explain that we switched to the human protein for the structural studies because the yeast protein was not soluble at the high concentrations needed for the biophysical studies.

Figure 7 (NMR data) requires more explanation to illuminate the differences between the WT and the mutant protein structure. As written, the information content is minimal.

We are uncertain what additional explanation is requested for the NMR data. As stated in the manuscript, we can conclude that the WT RBM28 RRM3 is a folded, globular protein based on the well-dispersed resonances in the _15_N-HSQC spectrum, and we can conclude that the L351P substitution disrupts the domain structure based on the clustering of resonances from 8.0 to 8.5 ppm in the _1_H dimension of the _15_N-HSQC spectrum. Additional analysis is included in the figure legend indicating that decreased dispersion of glutamine and asparagine side chain resonances are also consistent with disrupted domain structure.

Figure 7 needs labels: a) WT and mutant protein and b) location of the mutation L351P.

We have added labels to clarify that Figure 7 is a ribbon diagram of a homology model of the WT human RBM28 RRM3 with the amino acid that is mutated in ANE syndrome (L351) shown with red space-filling spheres. The two images are different views of the same model and do not represent the WT vs. the mutant protein, since the mutant protein is unstructured. We added a rotational arrow to make this clear.

Reviewer #4 (Additional data files and statistical comments):

As noted in the long form review, quantitative analyses are lacking in some places, and the statistical analyses performed in Figure 2 and Figure 6 need to be aligned.

We have included quantitative analyses of all western blots. We did not change the statistical analyses for the reasons stated above.